

# Regional Simulation of Indian summer Monsoon Intraseasonal Oscillations at Gray Zone Resolution

Xingchao Chen[1, 2, 3], Olivier M. Pauluis[1, 2], Fuqing Zhang[3]

[1]Center for Prototype Climate Modeling, New York University in Abu Dhabi, Abu Dhabi, United Arab Emirates

[2]Courant Institute of Mathematical Sciences, New York University, New York, New York, USA

[3]Department of Meteorology and Atmospheric Science, and Center for Advanced Data Assimilation and Predictability Techniques, The Pennsylvania State University, University Park, Pennsylvania, USA

Correspondence to:

Dr. Xingchao Chen
Email: xzc55@psu.edu



34          **Abstract**

36          Simulations of the Indian summer monsoon by cloud-permitting WRF model at gray zone

resolution are described in this study, with a particular emphasis on the model ability to capture
the Monsoon Intraseasonal Oscillations (MISO). Five boreal summers are simulated from 2007
to 2011 using the ERA-Interim reanalysis as lateral boundary forcing data. Our experimental
set-up relies on a high horizontal resolution of 9km to capture deep convection without the use of
a cumulus parameterization. When compared to simulations with coarser grid spacing (27-km)
and using the cumulus scheme, our approach results in a reduction of the biases in mean
precipitation and in more realistic reproduction of the low frequency variability associated with
MISO. Results show that the model at gray zone resolution captures the fundamental features of
the summer monsoon. The spatial distributions and temporal evolutions of monsoon rainfall in
WRF simulations are verified qualitatively well against observations from the Tropical Rainfall
Measurement Mission (TRMM), with regional maxima located over West Ghats, central India,
Himalaya foothills and the west coast of Myanmar. The onset, breaks and withdrawal of the
summer monsoon in each year are also realistically captured by the model. MISO phase
composites of monsoon rainfall, low-level wind and precipitable water anomalies in the
simulations are compared qualitatively with the observations. Both the simulations and
observations show a northeastward propagation of the MISO, with the intensification and
weakening of Somali Jet over the Arabian Sea during the active and break phases of the Indian
summer monsoon.





## 1. Introduction

The Indian summer monsoon (ISM) is the most vigorous weather phenomena affecting the Indian subcontinent every year from June through September (JJAS). It contributes about 80% of the total annual precipitation over the region (Jain and Kumar, 2012; Bollasina, 2014) and has substantial influences to the agricultural and industrial productions in India. The ISM exhibits strong low frequency variability in the form of "active" and "break" spells of monsoon rainfall (Goswami and Ajayamohan, 2001), with two dominant modes on timescales of 30-60 days (Yasunari, 1981; Sikka and Gadgil, 1980) and 10-20 days (Krishnamurti and Bhalme, 1976; Chatterjee and Goswami, 2004). The low-frequency mode is generally known as the Monsoon Intraseasonal Oscillation (MISO), which is closely related to the Boreal Summer Intraseasonal Oscillations (BSISO, Krishnamurthy and Shukla, 2007; Suhas et al., 2013; Sabeerali et al., 2017; Kikuchi et al., 2012; Lee et al., 2013) and characterized by a northeastward propagation of the precipitation from the Indian Ocean to the foothills of the Himalayan foothills  (Jiang et al., 2004). The MISO not only affects the seasonal mean strength of the ISM, but also plays a fundamental role in the interannual variability and predictability of the ISM (Goswami and Ajayamohan, 2001; Ajayamohan and Goswami, 2003). The MISO phases occurring at the early and late stages of the ISM also has considerable influences on the onset and withdrawal of the ISM, which, in another world, determining the length of the rainy season (Sabeerali et al., 2012). Hence, a more accurate forecast of the MISO assumes significance. The MISO is influenced by a number of physical processes (Goswami, 1994). Its interactions with the mean monsoon circulation and other tropical oscillations make its propagating characteristics more complex when compared with the eastward propagating Madden Julian oscillation (MJO, Madden and Julian, 1971).



General circulation models (GCMs) are broadly used to simulate the large-scale circulation
and seasonal rainfall climatology of the ISM. Results show that GCMs are able to capture the
fundamental features of the monsoon circulation reasonably well and also show some skills in
reproducing the seasonal-averaged distributions of the monsoonal rainfall (e.g., Bhaskaran et al.,
1995; Lau and Ploshay, 2009; Chen et al., 2011). However, the skill of the current generations of
GCMs in simulating and predicting the MISO remains poor (Ajayamohan et al., 2014; Lau and
Waliser, 2011). The computer power available nowadays constrains most GCMs to perform
long-term global simulations with a horizontal spacing lager than 100 km (Lucas-Picher et al.,
2011). As a result, the GCMs cannot well capture the high frequency atmospheric variance and
regional dynamics associated with the MSIO, which also leads to a systematic bias in simulating
the ISM rainfall (Goswami and Goswami, 2016; Srinivas et al., 2013). Increasing the spatial
resolution therefore is the way for GCMs (of course not the only way) to improve the MISO
simulation and to reduce the systematic model biases (e.g., Ramu et al., 2016; Rajendran and
Kitoh, 2008; Oouchi et al., 2009). However, the high resolution global simulations usually
require significant computational resources that most climate modeling groups cannot afford.
An alternative approach to improve the ISM and MISO simulations is the use of regional
climate models (RCMs). RCMs dynamically downscale the GCM simulations or reanalysis and
perform a climate simulation over a certain region of the globe (Prein et al., 2015; Giorgi, 2006).
Using same computer resources, RCMs are able to perform the climate simulation with much
higher spatial resolution and are expected to better capture the high pass atmospheric variance
and resolve the important regional forcings associated with topography, land-sea contrast and
land cover (Bhaskaran et al., 1996; Dash et al., 2006). Many previous studies found that better
ISM and MISO simulations can be achieved in the high resolution (typically 50km or less)





RCMs than that in the GCMs with coarser grid spacing (e.g., Bhaskaran et al., 1998; Kolusu et
al., 2014; Lucas-Picher et al., 2011; Srinivas et al., 2013; Raju et al., 2015; Samala et al., 2013;
Vernekar and Ji, 1999; Mukhopadhyay et al., 2010; Saeed et al., 2012). Nonetheless, apparent
biases of the MISO simulations can still be found in the most previous RCM studies. One
principle reason is, to reduce the computational requirements, the spatial resolutions used in the
previous RCM studies are still not high enough to resolve the convection explicitly, and
convective activity is represented by the cumulus parameterization schemes in the simulations.
However, the organization of convection is the primary mechanism for simulating the realistic
MISO (Ajayamohan et al., 2014). Hence, using cumulus schemes may introduce a systematic
bias in simulating the MISO and the monsoon rainfall climatology (Mukhopadhyay et al., 2010;
Das et al., 2001; Ratnam and Kumar, 2005). In addition, the cumulus parameterization schemes
can also interact with other parameterization schemes, such as the planetary boundary layer,
radiation and microphysical schemes, which may imply far-reaching consequences through
nonlinearities and affects the simulation of the MISO (Prein et al., 2015).
The alternative to the use of a convective parameterization is to rely on the internal dynamics
to resolve convective motion. A consensus view is that Cloud Resolving Models (CRMs) must
have a horizontal resolution of at least 2km to resolve the dynamics of deep convection, albeit
even finer resolution are necessary in order to adequately resolve the turbulent motions in
convective systems (Bryan et al., 2003). However, Pauluis and Garner (2006) have shown that
CRM with horizontal resolution as coarse as 12km can accurately reproduce the statistical
behavior of convection simulated at much finer resolution. This implies that a coarse resolution
CRM, one in which convective motion is under resolved, can nevertheless capture adequately the
impacts of convective motions on large scale atmospheric flows.



Recently, Wang et al. (2015, W15 hereafter) simulated two MJO events observed during the
CINDY/DYNAMO campaign using a convection-permitting regional model with 9-km grid
spacing. The authors compared the simulations with multiple observational datasets and found
that the RCM at this resolution can successfully capture the intraseasonal oscillations over the
tropical oceans. The horizontal grid spacing of 9 km used in W15 is not adequate for individual
convective cells, but enough to resolve the organized mesoscale convective systems and their
upscale impacts and coupling with large-scale dynamics. Hence, they called the 9-km grid
spacing as gray zone resolution in regional convection-permitting climate simulation. The
convection-permitting RCMs at the gray zone resolution have the twin advantages of (1) using
much less computational resources than that required by the typical cloud-resolving simulations
(usually, grid spacing should be smaller than 2 km) and (2) avoid using the cumulus
parameterization schemes. The primary objective of the present study is to evaluate the ISM and
MISO simulations in the RCM at the gray zone resolution, which could be an affordable and
efficient way for most climate model groups to achieve a cloud-permitting MISO simulation.
The paper is constructed as follows. Section 2 provides a brief description of the model and the
data used. Section 3 presents the model simulated mean ISM features and seasonal evolutions of
the rainfall over the monsoon region. The simulated MISO are described and compared with the
observations and reanalysis in section 4. Section 5 gives the concluding remarks of the study.

**2. Experimental setup and observational datasets**
The model configuration here is similar with the one used in W15. The Advance Research
WRF model (Skamarock et al., 2008), version 3.4.1, is used to simulate the ISMs over the Indian
subcontinent from 2007 to 2011. Simulations are performed over a single domain that covers the





most of South Asia with $777 \times 444$ grid points and 9-km grid spacing (Fig. 1). There are 45
vertical levels with a nominal top at 20 hPa and 9 levels in the lowest 1 km. Vertically
propagating gravity waves have been suppressed in the top 5 km of the model with the implicit
damping scheme (Klemp et al., 2008). The simulation employs the unified Noah land surface
physical scheme (Chen and Dudhia, 2001), the GCM version of the Rapid Radiative Transfer
Model (RRTMG) longwave radiation scheme (Iacono et al., 2008), the updated Goddard
shortwave scheme (Shi et al., 2010) and the WRF Double-Moment (WDM) microphysics
scheme (Lim and Hong, 2010) from WRF V3.5.1 with an update on the limit of the shape
parameters and terminal speed of snow. In W15, the authors used the Yonsei University (YSU)
boundary layer scheme (Hong et al., 2006) to simulate the subgrid-scale meteorological
processes within the planetary boundary layer. However, we find that there exists an apparent
dry bias in simulating the ISM precipitation after a long-term integration when YSU boundary
layer scheme is used. In order to improve the simulation, boundary layer scheme used for this
study has been changed to the new version of the asymmetric convective model (ACM2, Pleim,
2007). Hu et al. (2010) evaluated the different boundary schemes used in the WRF model and
found that ACM2 scheme can better simulate the boundary meteorological conditions of the
Texas region during summer than YSU scheme. Nevertheless, the sensitivity of ISM simulations
to the boundary-layer schemes is still deserve closer analysis and quantifications in the future,
which is out of the scope of the present study. Our model configuration does not use any
parameterization for deep convection, but rather relies on the internal dynamics to capture the
impact of convective activity.
Five boreal summers are simulated from 2007 to 2011 in this study. The 6-hourly
ERA-Interim reanalysis (Dee et al., 2011) is used as the initial and boundary conditions for the



simulations, and sea surface temperature (SST) is updated every 6 hours using the ERA-Interim
SST data. The model integrations start from 0000 UTC 20 April in each year. For the first 3 days,
a spectral nudging is applied to relax the horizontal wind with a meridional wave number 0-2 and
a zonal wavenumber 0-4, which constrains the large-scale flow and convergence in the domain
and allows the mesoscale to saturate in the spectral space (W15). The simulations are integrated
until October 30 for each year in order to capture the withdrawal of the ISM in different years.
The simulated spatial distributions and temporal variations of surface rainfall are verified against
the 3-hourly 0.25$^{o}$ TRMM 3B42 rainfall product version 7A, while the large-scale circulations
and atmospheric conditions in the simulations are verified against the ERA-interim reanalysis.
Besides the control simulations at 9 km resolution (WRF-gray hereafter), another set of
numerical simulations with coarser grid spacing (27km, WRF-27km hereafter) are also
conducted in this study to evaluate the extent to which the cloud-permitting simulations at gray
zone resolution can improve the simulation of the ISM and MISO. The configuration of the
coarse simulations is similar with WRF-gray except cumulus parameterization scheme is used to
represent the subgrid-scale convective activity. Mukhopadhyay et al. (2010) investigated the
impacts of different cumulus schemes on the systematic biases of ISM rainfall simulation in the
WRF RCM. They compared the simulations conducted with three different convective schemes,
namely the Grell–Devenyi (GD, Grell and Dévényi, 2002), the Betts–Miller–Janjić (BMJ, Janjić,
1994; Betts and Miller, 1986), and the Kain–Fritsch (KF, Kain, 2004) schemes. Results show
that KF has a high moist bias while GD shows a high dry bias in simulating the monsoonal
rainfall climatology. Among these three schemes, BMJ can produce the most reasonable
monsoonal precipitation over the Indian subcontinent with the least bias. Similar results can also



be found in Srinivas et al.(2013). Hence, the BMJ scheme has been used in the WRF-27km
simulations.
Fig. 2 shows the daily surface precipitation averaged over the Indian subcontinent (shown by
the blue polygon in Fig. 1) from TRMM observation, WRF-gray and WRF-27km during the
monsoon seasons (JJAS). An apparent moist bias of surface precipitation can be found for all 5
years (2007 to 2011) in WRF-27km, while this systematic bias is reduced considerably in
WRF-gray. In addition, we can find that the simulations at gray zone resolution (WRF-gray) can
better capture the interannual variability of the monsoon rainfall amount than the coarse WRF
simulations (WRF-27km). Beside the Indian subcontinent, WRF-27km also shows high moist
biases of surface rainfall over the adjacent oceans to the west and east coasts of India and
Himalaya foothills. Similar results can also be found in the earlier studies (e.g., Srinivas et al.,
2013). While, this moist biases over oceanic and mountainous area are reduced dramatically in
WRF-gray (not shown here). Results show that the monsoonal rainfall climatology can be better
simulated in the cloud-permitting RCM at gray zone resolution than that in the RCM using the
convective parameterization schemes. The rest of this paper will focus on the assessment of the
ISM and MISO simulations in WRF-gray while both the MISO simulations in WRF-gray and
WRF-27km will be compared to the observations in section 4.

**3. Mean features of Indian Summer Monsoon**
The large-scale atmospheric circulation and temporal-spatial patterns of the monsoon rainfall
in WRF-gray are first assessed in this section. Fig. 3a and 3b show the 5-yr JJAS climatological
mean 200-hPa winds and geopotential heights extracted from ERA-Interim and WRF-gray.
During the summer monsoon, the upper troposphere (200 hPa) is characterized by a strong



anti-cyclone over the Tibetan plateau and easterly winds over the Indian subcontinent. The
model well captures the wind pattern and geopotential height in the upper troposphere, though
the Tibetan high-pressure and easterly winds in WRF-gray are slightly stronger than that in
ERA-Interim (Fig. 3b). At lower level (850-hPa), the model realistically simulates the
geographical position and strength of Somali Jet over the Arabian Sea, with a slight
overestimation of the wind speed (Fig. 3c and 3d). Moisture is transported by the strong
low-level winds from the Arabian Sea to the Indian subcontinent. As a result, a precipitable
water maximum can be found over West Ghats and the Eastern coast of the Arabian Sea in both
in ERA-Interim and WRF-gray, though the precipitable water over the mountainous rages of
West Ghats in WRF-gray is a little higher than that in ERA-Interim. In addition, WRF-gray also
well captures the rain shadow downwind of the mountainous areas of central and southern India
where a slight dry bias can be noticed (Fig. 3d). The low-level southwesterly winds over the Bay
of Bengal in WRF-gray are stronger than that in ERA-Interim, which leads to an overestimation
in the precipitable water over the north tip of the Bay of Bengal, the west coast of Myanmar and
the foothills of Himalaya (Fig. 3d). A comparison of JJAS-averaged daily rainfall distribution
observed by TRMM with that simulated by WRF-gray is shown in Figs 3e and 3f. In general,
WRF-gray realistically captures the spatial pattern of the monsoon rainfall with the regional
rainfall maximums over West Ghats, central India, Himalaya foothills and the west coast of
Myanmar. Consistent with the biases shown in the low-level wind and precipitable water fields
(Fig. 3d), the simulated surface rainfall shows a dry bias over central India and a moist bias over
West Ghats, Himalaya foothills and the west coast of Myanmar (the Bay of Bengal). Similar
features can also been found in earlier RCM studies (e.g., Lucas-Picher et al., 2011; Rockel and
Geyer, 2008), which have shown that these biases can be explained by the way that surface





schemes cannot well simulate the land-sea pressure and temperature contrasts that driving the
monsoon dynamics and induce an overestimation of surface wind speed over oceans. This results
in an overestimation of the surface evaporation over the tropical oceans and excess precipitation
downstream over the mountain ranges of South-East Asia.
The Somali Jet over the Arabian Sea is a central figure of the Indian Summer Monsoon. Its
emergence is crucial in determining the onset precipitation over the Indian subcontinent (Ji and
Vernekar, 1997; Joseph and Sijikumar, 2004). Ajayamohan (2007) proposed an index to
represent the Kinetic Energy (KE) of Somali Jet (KELLJ), which is defined as the mean KE of
winds at 850 hPa averaged over $50^{o}$-$65^{o}$E and $5^{o}$-$15^{o}$N (shown by the black box in Fig. 1). The
same index is applied here to assess the strength of Somali Jet. The 5-yr temporal evolutions of
KELLJ calculated from WRF-gray are compared with that calculated from ERA-Interim in Fig.
4. In general, the model well captures the evolution of KELLJ in different years. Sudden
increases in KE of Somali Jet in late May associated with the monsoon onsets are well
reproduced in WRF-gray. The Somali Jet is stronger during the monsoon (JJAS) than in May and
October, which leads to a stronger precipitation over the Indian subcontinent during the ISM.
WRF-gray also well simulates the intraseasonal variation of KELLJ and the decrease of KE
associated with the withdrawal of the monsoon in each year. Overall, the strength of Somali Jet
in WRF-gray is slightly stronger than that in ERA-Interim, which is similar with the above
analysis of Figs. 3c and 3d.
The evolution of surface rainfall averaged over the Indian subcontinent (shown by the blue
polygon) from WRF-gray is compared with that from TRMM observations (Fig. 5). Generally
speaking, WRF-gray well captures the mean strength and intraseasonal variation of the monsoon
rainfall. In these 5 years, the accumulated monsoonal rainfall amount over the Indian



subcontinent is largest in 2007 and smallest in 2009. 2009 is also one of the most drought years
in the past 3 decades. Corresponding to the evolution of Somali Jet, rainfall over the Indian
subcontinent begins to increase from late May, reaches it maximum during JJAS and decreases
again in late September or early October, which are associated with the onsets and withdrawals
of the ISM. The onset and withdrawal of the ISM are well captured by WRF-gray in most years
except the onset of the 2007 ISM in WRF-gray is later than that in TRMM observations. The
main reason of the 2007 ISM later onset in WRF-gray is that the super cyclonic storm Gonu
which induced strong precipitation over the west India and had considerable influence on the
onset of the 2007 ISM (Najar and Salvekar, 2010) has not been well captured in the WRF
simulation (the position of Gonu has a southwest shift in WRF-gray, not shown here). The ISM
also shows a strong ISO in each year in the form of "active" and "break" spells of monsoon
rainfall over the Indian subcontinent. These "active" and "break" phases of ISM are closely
related to the strengthening and weakening of Somali Jet (Fig. 4). Despite the biases of the
monsoon rainfall intensity, we can find that WRF-gray well captures most "active" and "break"
spells of 5-yr ISMs, which gives us confidence that the MISO can be qualitatively simulated in
the RCMs at gray zone resolution.
The spatial distributions of monthly mean precipitation from TRMM and WRF-gray in 2007,
2009 and 2011 are compared in Figs 6, 7 and 8. Similar with the analysis of Fig.3, the model
well captures the rainfall centers over West Ghats, central Indian, Himalaya foothills and the
west coast of Myanmar during the summer monsoon seasons, with an overestimation of
precipitation over the west coast of Myanmar and Himalaya foothills due to the overprediction of
low-level wind over the Bay of Bangle. With high spatial resolution, WRF-gray is able to
capture the finer details of orographic precipitation over the mountainous rages (for example,



along the west coastline of the Indian subcontinent). In addition, the interannual variability of
monsoon rainfall is also well simulated in WRF-gray (Figs. 6, 7 and 8). In 2007, rainfall is very
weak over the Indian subcontinent in May though orographic precipitation can still be found over
the mountainous rages along the west coastline (Figs. 6a and 6g). Accompany with the onset of
the ISM and the enhancement of low-level winds over the Arabian Sea, precipitation over the
west coast of Indian subcontinent and its adjacent oceans increases dramatically in June (Figs. 6b
and 6h). In July, the precipitation center along the west coast of Indian subcontinent is still
apparent and the precipitation over central India is increased considerably (Figs. 6c and 6i).
Rainfall over Himalaya foothills and the west coast of Myanmar also reaches its strongest stage
in this month. In August, rainfall over central India and the west coast of Myanmar are still
strong while the precipitation near the Himalaya foothills is decreased (Figs. 6d and 6j). The
rainfall intensity over the entire monsoon region decreases continually in September (Figs. 6e
and 6k) and the precipitation over the Indian subcontinent becomes very weak in October (Figs.
6f and 6l), which represents the end of the monsoon season. When compared to 2007, the ISM in
2009 is drier, especially over the Indian subcontinent (Fig. 7). The onset and withdrawal of the
2009 ISM over the Indian subcontinent are in June and September. The significant "break" spells
of the 2009 ISM in June, August and September are well captured by WRF-gray (Figs. 5c, 7h, 7j
and 7k). The evolution of monthly mean precipitation in 2011 (Fig. 8) is similar with that in
2007 (Fig. 6) with the rainfall over the central India reaches its strongest stage in August (Figs.
8d and 8j). In May 2011, an apparent moist bias of precipitation can be found over the Arabian
Sea in WRF-gray, which is induced by the formation of an unreal tropical cyclone in the
simulation. Generally speaking, WRF-gray is able to capture the spatial and temporal features of
the ISM rainfall. Though apparent biases can still be found, the intensity and spatial pattern of





monsoon rainfall in WRF-gray are verified well against the observations, especially over the
Indian subcontinent.

## 4. Monsoon Intraseasonal Oscillations (MISO)

As mentioned in the Introduction, the MISO has fundamental influences on the seasonal
mean, predictability and interannual variability of the ISM. Hence, the simulation of the MISO is
very important for the credibility of the model in simulating the ISM. The section evaluates the
ability of WRF-gray in simulating the MISO. MISO Phase composites of the surface rainfall and
large-scale flows from WRF-gray are compared with that from the observations

### 4.1 Indices for the MISO

Using the developed nonlinear Laplacian spectral analysis (NLSA) technique (Giannakis and
Majda, 2012a, b), Sabeerali et al. (2017) developed improved indices for real-time monitoring of
the MISO. Compared to the classical covariance-based approaches (for example Suhas et al.,
2013), a key advantage of NLSA is that it is able to extract the spatiotemporal modes of
variability spanning multiple timescales without requiring bandpass filtering or seasonal
partitioning of the input data. The MISO indices constructed by NLSA better resolve the
temporal and spatial characteristics of the MISO when compared to the conventional
EEOF-based MISO indices. In order to evaluate the MISO simulation in WRF-gray, the NLSA
MISO indices are applied in this study to construct the phase composites of rainfall and
atmospheric circulation from WRF-gray and the observations. Fig. 9 shows the daily evolution
of the MISO in each year monitored by the two-dimensional phase space diagram constructed
from the NLSA MISO indices. As with Sabeerali et al. (2017), all indices are extracted from the





daily GPCP rainfall dataset (Huffman et al., 2001). The 2D phase space of the NLSA MISO
indices is divided into 8 phases to represent different phases of the MISO. The significant MISO
event is defined as the instantaneous MISO whose amplitude is greater than 1.5 (shown by the
black circle in Fig. 9). From Fig. 9, we can find that the MISO activity in 2007, 2008 and 2009
are much more significant than that in 2010 and 2011. Among these five years, the accumulated
monsoon rainfall amount over the Indian subcontinent is highest in 2007, which also features the
strongest MISO activity (Fig. 9a). The following year, 2008, is also a moist year with strong
MISO activity is from mid-June to the end of September (Fig. 9b). In 2009 (Fig. 9c), a severe
drought year, the MISO is weak during the early and late stages of the monsoon season (June and
September), but stronger in the midst of the monsoon season (July and August). The amplitude
of the MISO indices in 2010 and 2011 are much smaller, while significant MISO events can still
be found in September 2010 (Fig. 9d) and most monsoon months in 2011 (Fig. 9e).

**4.2 Phase composites of surface rainfall**
Fig. 10 shows the phase composites of daily surface rainfall anomalies (subtracted the mean
daily rainfall of 5-yr monsoon seasons) obtained from TRMM observation based on the NLSA
MISO indices. The phase composites are computed by averaging the significant MISO activities
in each phase space occurred in the 5-yr monsoon seasons. An apparent northeastward
propagation of the MISO can be found in the phase composites (from the phase 1 to the phase 8),
which corresponds to the anticlockwise rotation in the 2D phase space of the MISO indices (Fig.
9). Phase 1 shows the formation of enhanced rainfall anomalies over the tropic Indian Ocean (Fig.
10a). During this Phase, rainfall over the Indian subcontinent is suppressed. The enhanced
rainfall anomalies over the tropic ocean become stronger and move toward the Indian





353 subcontinent in Phase 2 (Fig. 10b) and reach West Ghats and its adjoining oceans in Phase 3 (Fig.

354 10c). In Phase 3, precipitation over the Indian subcontinent is enhanced while rainfall over the

355 Bay of Bengal is suppressed (Fig. 10c). Rainfall over central India and the south part of the Bay

356 of Bengal are enhanced considerably in Phase 4 (Fig. 10d) and form into a northwest-southeast

357 enhanced rainfall line that stretches from the west coast of the Indian subcontinent to the south of

358 the Indochina in Phase 5 (Fig. 10e). This enhanced rainfall line continually propagates to

359 northeast in Phase 6 (Fig. 10f) and collapses in Phase 7 (Fig. 10g). In Phase 7, the enhanced

360 rainfall anomalies can still be found over north India while the rainfall in south India is

361 suppressed by the MISO. The total rainfall over the entire basin is weakest during Phase 8 with

362 the rainfall anomalies are mostly negative over the inland regions of India (Fig. 10h). However,

363 rainfall near Himalaya foothills begins to increase in this phase and reaches its maximum in

364 Phase 2 (Fig. 10b). The phase composites of daily surface rainfall anomalies obtained from 5-yr

365 TRMM observations in this study are similar to the 26-yr phase composites in Sabeerali et al.

366 (2017), which shows that the 5-yr rainfall statistic reflects the climatological characteristics of

367 the MISO.

368 Fig.11 presents the phase composites of daily surface rainfall anomalies obtained from

369 WRF-gray. Despite differences in the intensity and location of rainfall anomalies, the MISO

370 simulation in WRF-gray verified well against the TRMM observations. The fundamental features

371 of rainfall anomalies in all 8 Phases of the MISO are well captured by WRF-gray: for example,

372 the northeastward propagation of the enhanced rainfall anomalies, the "active" and "break"

373 phases of the monsoon rainfall over the Indian subcontinent, the northwest-southeast enhanced

374 rainfall line in Phases 5 and 6, the increase of rainfall over Himalaya foothills from Phase 8 to

375 Phase 2 and so on. Nonetheless, we should also notice that the amplitude of the rainfall





anomalies in WRF-gray is slightly larger than that in the TRMM observations, which reflects

that the model simulated MISO is stronger than the real one in the satellite observations.

In order to evaluate to what extent the RCM at gray zone resolution can improve the

simulation of the MISO, the phase composites of daily surface rainfall anomalies obtained from

WRF-27km (Fig. 12) are also compared with that from the TRMM observations (Fig. 10) and

WRF-gray (Fig. 11) in this section. We can find that the amplitude of rainfall anomalies in

WRF-27km is much larger than that in WRF-gray and TRMM observations, which shows the

WRF-27km has larger systematic biases than WRF-gray in simulating the MISO intensity.

Though WRF-27km can also basically capture the "active" (Figs 12c, 12d, 12e and 12f) and

"break" (Figs. 12g, 12h, 12a and 12b) phases of the ISM, it shows a larger bias in the

spatial-temporal distributions of the rainfall anomalies during the different phases of the MISO

than WRF-gray. For example, the rainfall anomalies in Phase 1, 2 and 3 (Figs. 12a, 12b and 12c)

are shifted northward, consistent with a faster development of the MISO cycle in the coarse

resolution model. The northwest-southeast enhanced rainfall line shown in TRMM observations

and WRF-gray is not clear in WRF-27km. This could be possibly due to deficiencies in how

WRF-27km capture stratiform rainfall, which would create a bias toward more patchy, deep

convective events. The increase of rainfall over Himalaya foothills from Phase 8 to Phase 2 has

not been well simulated in WRF-27km. Generally speaking, WRF-gray better simulates the

MISO than WRF-27km, both in the aspects of intensity and the spatial-temporal evolution.

Besides the phase composite, the evolutions of 10-day averaged daily surface rainfall

anomalies in WRF-gray and TRMM observations are also compared with each other to further

access the credibility of WRF-gray in simulating the intraseasonal variability of the ISM. 10-day

evolutions of rainfall anomalies from 1 July to 10 August, 2009 in TRMM observations and





WRF-gray are shown in Fig. 13. During this period, the monsoon rainfall over the Indian
subcontinent turns from a strong "active" phase to a strong "break" phase (Fig. 5c). The rainfall
is enhanced over the west coast of the Indian subcontinent, central India and the Bay of Bengal
in the first ten days of July (Fig. 7a), which is similar with the combined features of Phases 3 and
4 (Figs. 10c and 10d). The enhanced rainfall anomalies form into a northwest-southeast line in
the middle of July (Fig. 7b), which corresponds to Phases 5 and 6. In the end of July, rainfall
over most area of the Indian subcontinent is suppressed while the rainfall anomalies over north
India is still positive (Fig. 7c). In early August, rainfall anomalies over the entire Indian
subcontinent turn to negative with rainfall over Himalaya foothills is enhanced (Fig. 7d), which
is similar to the combined features of Phases 8, 1 and 2 (Figs. 10h, 10a and 10b). Though small
biases can be found in the simulated rainfall intensity and location, the 10-day evolutions of daily
rainfall anomalies in WRF-gray verified well against the TRMM observations (Figs. 13e-h),
which again proves that the cloud-permitting RCM at gray zone resolution is credible in
simulating the MISO.

**4.3 Phase composites of atmospheric circulation**

415        During the different phases of the MISO, the large-scale flows and atmospheric conditions

also exhibit different behaviors (Raju et al., 2015; Goswami et al., 2003; Mukhopadhyay et al.,
2010). Fig. 14 shows the phase composites of 850-hPa wind and precipitable water anomalies
obtained from ERA-Interim. Consistent with the phase evolution of the enhanced daily rainfall
anomalies (Fig. 10), the precipitable water anomalies also show an apparent northeastward
propagation from Phase 1 (Fig. 14a) to Phase 8 (Fig. 14h), which corresponds to the
anticlockwise rotation in the 2D phase space of the MISO indices (Fig. 9). The major features of





the MISO active phase (Fig. 14e) are the formation of low pressure anomalies over northwest
and central India which is associated with the southward shifting of monsoon trough (Raju et al.,
2015). As a result, the strong westerly wind over the Arabian Sea and the Bay of Bengal also
enhanced dramatically during the active phase of the MISO, which transports more water vapor
from the oceans to the inland regions and leads to enhanced precipitable water anomalies over
the land. The strength of Somali Jet is also enhanced during the MISO active phase (Fig. 14e).
During the break phase of the MISO, on the other hand, high pressure anomalies can be found
over northwest and central India, which is associated with the northward shifting of monsoon
trough. The westerly wind over the Arabian Sea and Somali Jet are weakened during the break
phase (Figs. 14a and 14b), which lead to negative precipitation water and surface rainfall
anomalies over the Indian subcontinent. Fig. 15 shows the phase composites of 850-hPa wind
anomalies and precipitable water anomalies obtained from WRF-gray. We can find that
WRF-gray well produces the features of large-scale flow and precipitable water anomalies in
different phases of the MISO (Fig. 15), which shows that the cloud-permitting RCM at gray zone
resolution can also well capture the large-scale circulation features of the MISO. We should
notice that, as the rainfall anomalies shown in Fig. 11, the amplitudes of low-level wind and
precipitable water anomalies in WRF-gray (Fig. 15) are larger than that in ERA-Interim (Fig. 14),
which implies that the simulated MISO in WRF-gray is stronger than the real one.

**4.4 Sensitivity to initial dates**

442       While WRF-gray captures many aspects of the ISM and MISO qualitatively, quantitative

model biases are still apparent. These biases may be induced by various reasons such as the
choices of surface scheme which may induce biases in simulating the surface temperature and





land-sea contrast, the model domain size which have significant effects on the simulation of
regional features and the initial conditions which the dynamical systems may be highly sensitive
to. The sensitivity of WRF-gray simulation to initial dates is further investigated in this section.
Fig. 16 shows the temporal evolutions of Somali Jet Strength (Fig. 16a), precipitation water (Fig.
16b) and precipitation (Fig. 16c) averaged over the Indian subcontinent in the WRF simulations
at gray zone resolution started from three different days (WRF0420: blue lines, started from
0000 UTC 20 April; WRF0419: red lines, started from 0000 UTC 19 April; WRF0421: green
lines, started from 0000 UTC 21 April) in 2007. Though all three WRF simulations are forced by
the same lateral boundary conditions and the initial times are also close to each other, we can still
find apparent differences of the simulated monsoon atmospheric circulation (Fig. 16a), humidity
(Fig. 16b) and precipitation (Fig. 16c) in three experiments. In particular, in May, there exist
apparent rainfall biases in WRF0419. However the onset of the ISM is better captured by
WRF0419 than WRF0420 and WRF0421. The overprediction of monsoon rainfall from 15
September to 01 October in WRF0420 is considerably reduced in WRF0419 and WRF0421.
Results show that the ISM and MISO simulations in RCM at gray zone resolution are sensitive to
the initial conditions.

**5. Summary and discussion**

463       Simulations of the ISM by cloud-permitting WRF model at gray resolution (9 km) are

evaluated in this study, with a particular emphasis on the credibility of the MISO simulation. The
model is forced by ERA-Interim reanalysis for every year from 20 April to 30 October during
2007-2011. Model domain covers the entire Indian monsoon region which allows the systematic
evolution of the ISM internal dynamics. Compared with the RCM at coarse resolution and using



the cumulus parameterization scheme (WRF-27km), the systematic biases of monsoon rainfall
climatology in the cloud-permitting RCM at gray zone resolution (WRF-gray) are reduced
considerably. The interannual variability of the accumulated monsoon rainfall over the Indian
subcontinent is also better captured in WRF-gray.
Results from WRF-gray are compared quantitatively with the reanalysis and long-term TRMM
observations. In general, WRF-gray could reproduce the fundamental features of ISM reasonably
well. The Tibetan high-pressure and easterly winds at 200 hPa in WRF-gray are slightly stronger
than that in ERA-Interim. The low-level southwesterly winds over the Bay of Bengal in
WRF-gray is also stronger when compared to that in the reanalysis, which leads to an
overprediction of precipitable water and surface rainfall over the west coast of Myanmar and
Himalaya foothills in WRF-gray. The temporal evolutions of Somali jet and surface rainfall
averaged over the Indian subcontinent are also well simulated in WRF-gray. The model captures
most onsets, breaks and withdrawals of the ISMs, while the ISM onset in 2007 is later in
WRF-gray than that in TRMM observation. Spatial distributions of monthly mean precipitation
from TRMM and WRF-gray are further compared in the current study. Results show that
WRF-gray could reproduce the spatial patterns of the monthly rainfall in each year and well
capture the monsoon rainfall centers over West Ghats, central India, Himalaya foothills and the
west coast of Myanmar. However, biases of rainfall intensity and position can still be found in
WRF-gray, for example, the model simulates an unreal tropical cyclone over the Arabian Sea in
May 2011.
Because the MISO has fundamental influences on the simulation and prediction of the ISM,
the skill of WRF-gray in simulating the MISO is quantitatively assessed in this study. The NLSA
MISO indices developed by Sabeerali et al. (2017) are applied in this study to construct the



MISO phase composites of surface rainfall and atmospheric circulations from WRF-gray and
observations. The enhanced rainfall anomalies show a clear northeastward propagation from the
MISO Phases 1 to 8. WRF-gray well captures this northeastward propagation and also simulates
the spatial distribution of rainfall anomalies during different phases of the MISO. The low-level
westerly wind over the Arabian Sea and Somali jet are strengthened (weakened) during the
active (break) phase of the MISO, which induces higher (lower) precipitable water and stronger
(weaker) precipitation over the Indian subcontinent. These features can also be well reproduced
in WRF-gray, though the amplitude of rainfall, precipitable water and wind anomalies in
WRF-gray are larger than that in observations. When compared with WRF-27km, the systematic
biases in simulating the MISO have been reduced considerably in WRF-gray, which shows that
the cloud-permitting RCM is able to improve the simulations of the MISO associated with the
ISM.
While WRF-gray captures many aspects of the ISM and MISO qualitatively, quantitative
model biases are still apparent. These biases may be induced by various reasons such as the
initial conditions. More comprehensive investigation of the predictability of the ISO and MISO
in RCM at gray zone resolution is deserved future studies.



**Acknowledgements**: Many thanks to Ajaya Ravindran and Sabeerali Cherumadanakadan
Thelliyil for the multiple discussions that led to this paper. The author Xingchao Chen and
Olivier Pauluis are supported by the New York University in Abu Dhabi Research Institute under
grant G1102. The computations were carried out on the High Performance Computing resources



at NYUAD. TRMM precipitation data were obtained from the NASA Goddard Space Flight
Center. ECMWF reanalysis data were retrieved from the ECMWF Public Datasets web interface
(http://apps.ecmwf.int/datasets/). WRF output can be made accessible by contacting
xzc55@psu.edu.





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



685                                          Figures

Figure 1. Model domain used in the WRF simulations with topography (gray scales) and
coastlines (red lines). The black box shows the climatic zone used for the calculation of KELLF
index and the blue polygon shows the Indian subcontinent.
Figure 2. Averaged daily rainfall over the Indian subcontinent for JJAS in different years from
TRMM observation (blue bars), WRF-gray (green bars) and WRF-27km (yellow bars).
Figure 3. 5-yr mean monsoon (JJAS) winds (vectors) and geopotential heights (red contours)
at 200-hPa from (a) ERA-Interim and (b) WRF-gray; winds (vectors) and precipitable water
(color shadings) at 850-hPa from (c) ERA-Interim and (d) WRF-gray; daily surface precipitation
(color shadings) from (e) TRMM and (f) WRF-gray. Topography is shown by the black contours
starts at 500m with a 1000-m interval.
Figure 4. Temporal evolution of KELLF indices in (a) 2007; (b) 2008; (c) 2009; (d) 2010 and
(e) 2011 from ERA-Interim (black lines) and WRF-gray (blue lines). A 5-day moving average is
applied to the time series.
Figure 5. Temporal evolution of daily surface rainfall averaged over the Indian subcontinent in
(a) 2007; (b) 2008; (c) 2009; (d) 2010 and (e) 2011 from TRMM (black lines) and WRF-gray
(blue lines). A 5-day moving average is applied to the time series.
Figure 6. Spatial distributions of averaged daily surface precipitation from May to October in
year 2007 derived from (a-f) TRMM and (g-l) WRF-gray.
Figure 7. Spatial distributions of averaged daily surface precipitation from May to October in
year 2009 derived from (a-f) TRMM and (g-l) WRF-gray.
Figure 8. Spatial distributions of averaged daily surface precipitation from May to October in
year 2011 derived from (a-f) TRMM and (g-l) WRF-gray.





Figure 9. 2D phase space diagrams for the NLSA MISO indices. An anticlockwise propagation from the phase 1 represents MISO's northward propagation. The circle centered at the origin has radius equal to 1.5, which is the threshold for identification of significant MISO events.

Figure 10. Phase composites of daily surface rainfall anomalies obtained from TRMM (Figure a-h: phase 1 to 8).

Figure 11. Phase composites of daily surface rainfall anomalies obtained from WRF-gray (Figure a-h: phase 1 to 8).

Figure 12. Phase composites of daily surface rainfall anomalies obtained from WRF-27km (Figure a-h: phase 1 to 8).

Figure 13. Spatial distributions of 10-day averaged daily surface rainfall anomalies in (a, e) 1-10 July, (d, f) 11-20 July, (c, g) 21-31 July and (d, h) 01-10 August, 2009 derived from TRMM (left panels) and WRF-gray (right panels).

Figure 14. Phase composites of 850-hPa wind and precipitable water anomalies obtained from ERA-Interim (Figure a-h: phase 1 to 8).

Figure 15. Phase composites of 850-hPa wind and precipitable water anomalies obtained from WRF-gray (Figure a-h: phase 1 to 8).

Figure 16. Temporal evolutions of (a) KELLF indices, (b) precipitable water averaged over the Indian subcontinent and (c) daily surface precipitation averaged over the Indian subcontinent in year 2007 from ERA-Interim/TRMM (black lines), WRF-gray simulation starts from April 20 (blue lines, control run), WRF-gray simulation starts from April 19 (red lines) and WRF-gray simulation starts from April 21 (green lines).



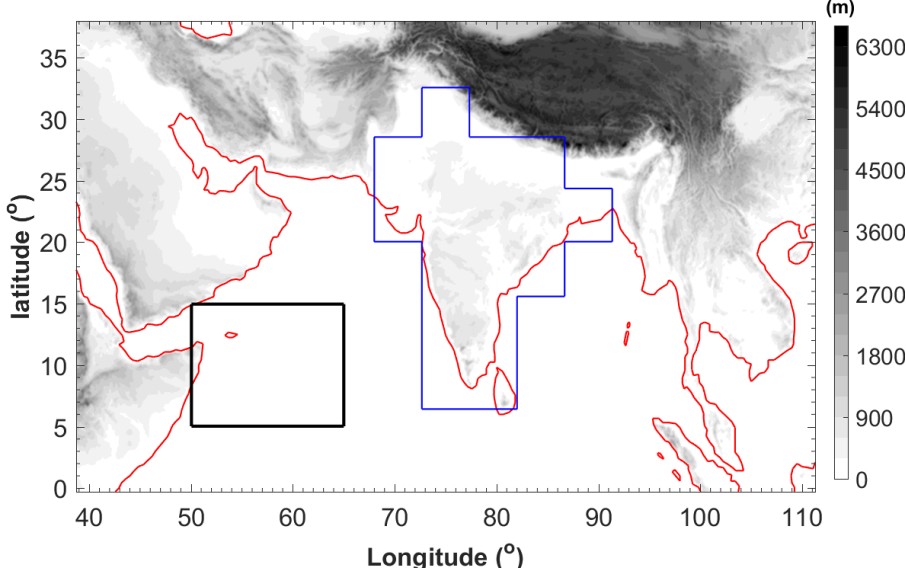

Figure 1. Model domain used in the WRF simulations with topography (gray scales) and coastlines (red lines).
The black box shows the climatic zone used for the calculation of KELLF index and the blue polygon shows the
Indian subcontinent.



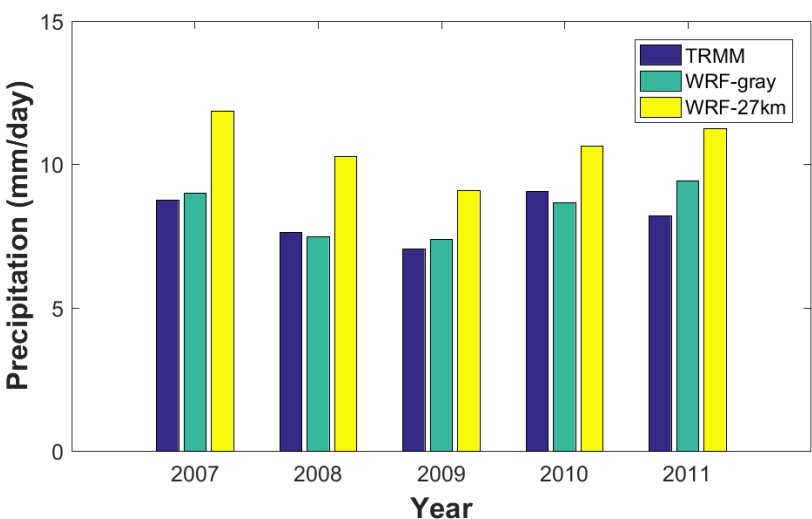

Figure 2. Averaged daily rainfall over the Indian subcontinent for JJAS in different years from TRMM
observation (blue bars), WRF-gray (green bars) and WRF-27km (yellow bars).





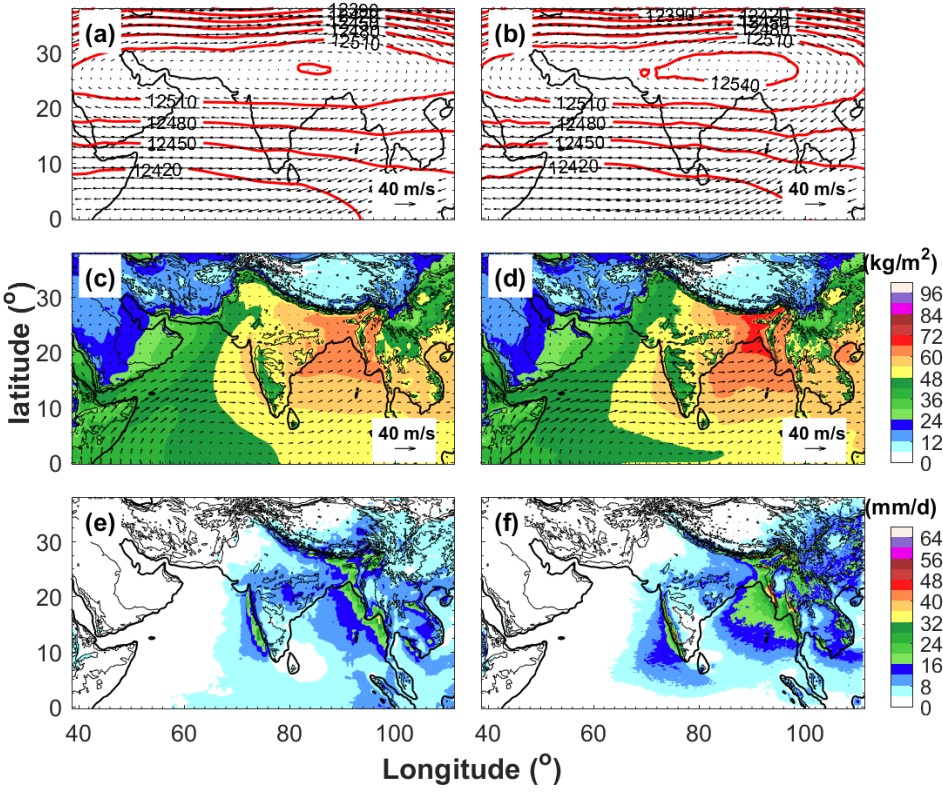

Figure 3. 5-yr mean monsoon (JJAS) winds (vectors) and geopotential heights (red contours) at 200-hPa from
(a) ERA-Interim and (b) WRF-gray; winds (vectors) and precipitable water (color shadings) at 850-hPa from (c)
ERA-Interim and (d) WRF-gray; daily surface precipitation (color shadings) from (e) TRMM and (f) WRF-gray.
Topography is shown by the black contours starts at 500m with a 1000-m interval.





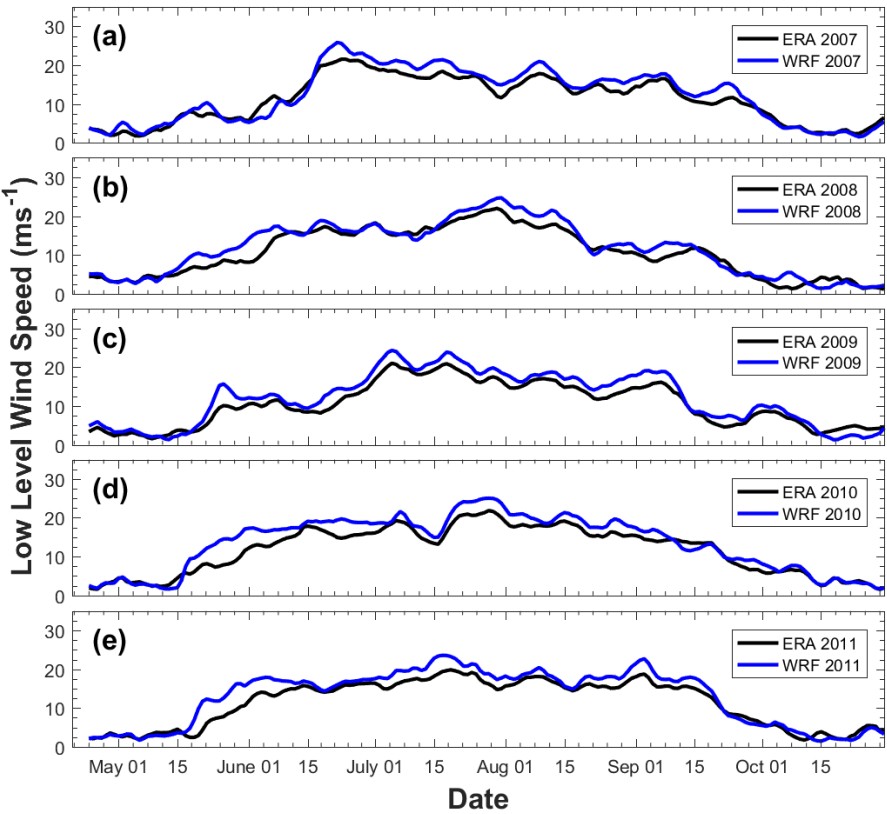

Figure 4. Temporal evolution of KELLF indices in (a) 2007; (b) 2008; (c) 2009; (d) 2010 and (e) 2011 from
ERA-Interim (black lines) and WRF-gray (blue lines). A 5-day moving average is applied to the time series.




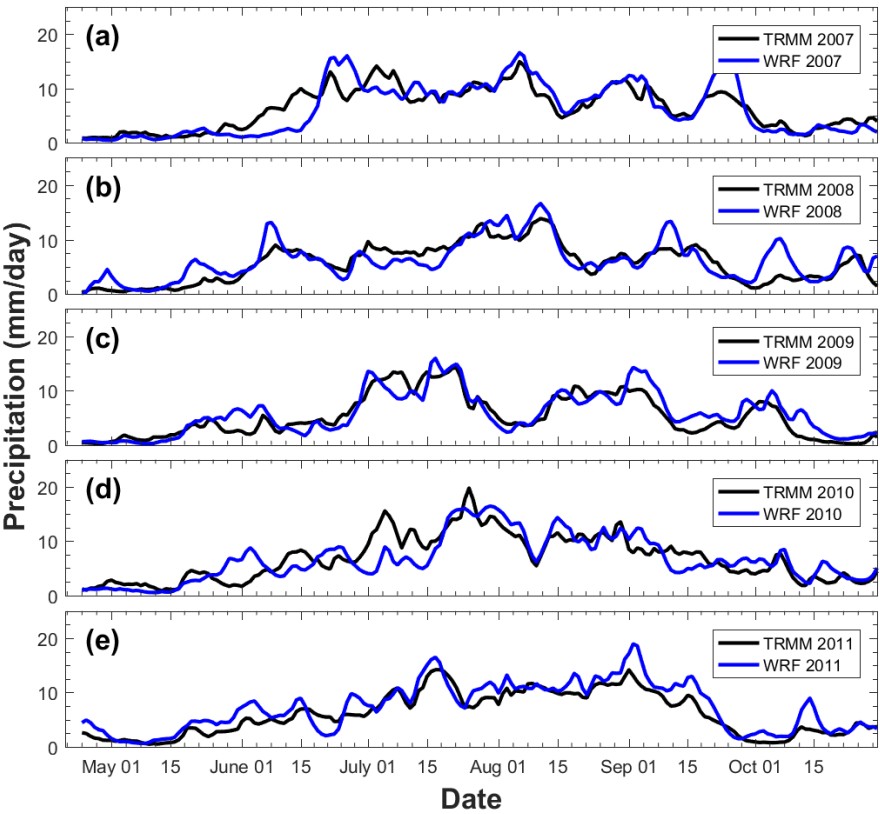

Figure 5. Temporal evolution of daily surface rainfall averaged over the Indian subcontinent in (a) 2007; (b)
2008; (c) 2009; (d) 2010 and (e) 2011 from TRMM (black lines) and WRF-gray (blue lines). A 5-day moving
average is applied to the time series.





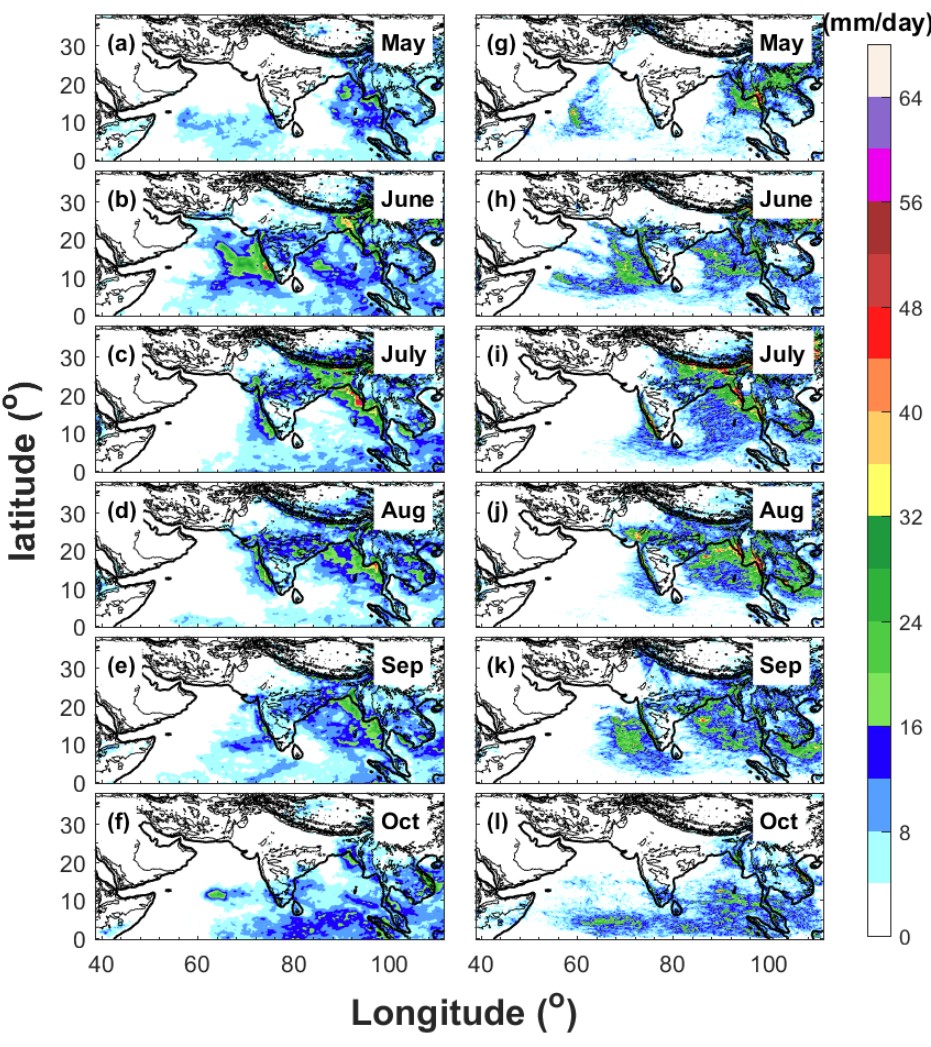

Figure 6. Spatial distributions of averaged daily surface precipitation from May to October in year 2007 derived from (a-f) TRMM and (g-l) WRF-gray.



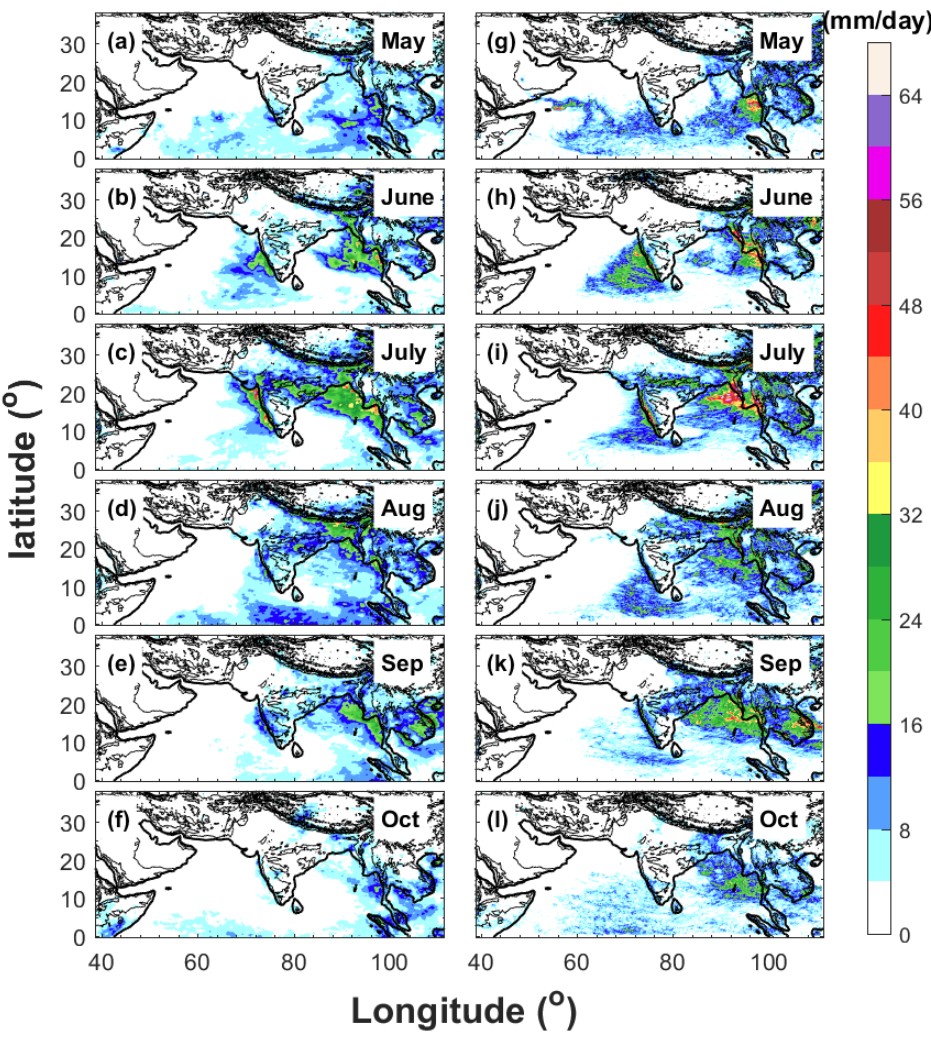

Figure 7. Spatial distributions of averaged daily surface precipitation from May to October in year 2009
derived from (a-f) TRMM and (g-l) WRF-gray.



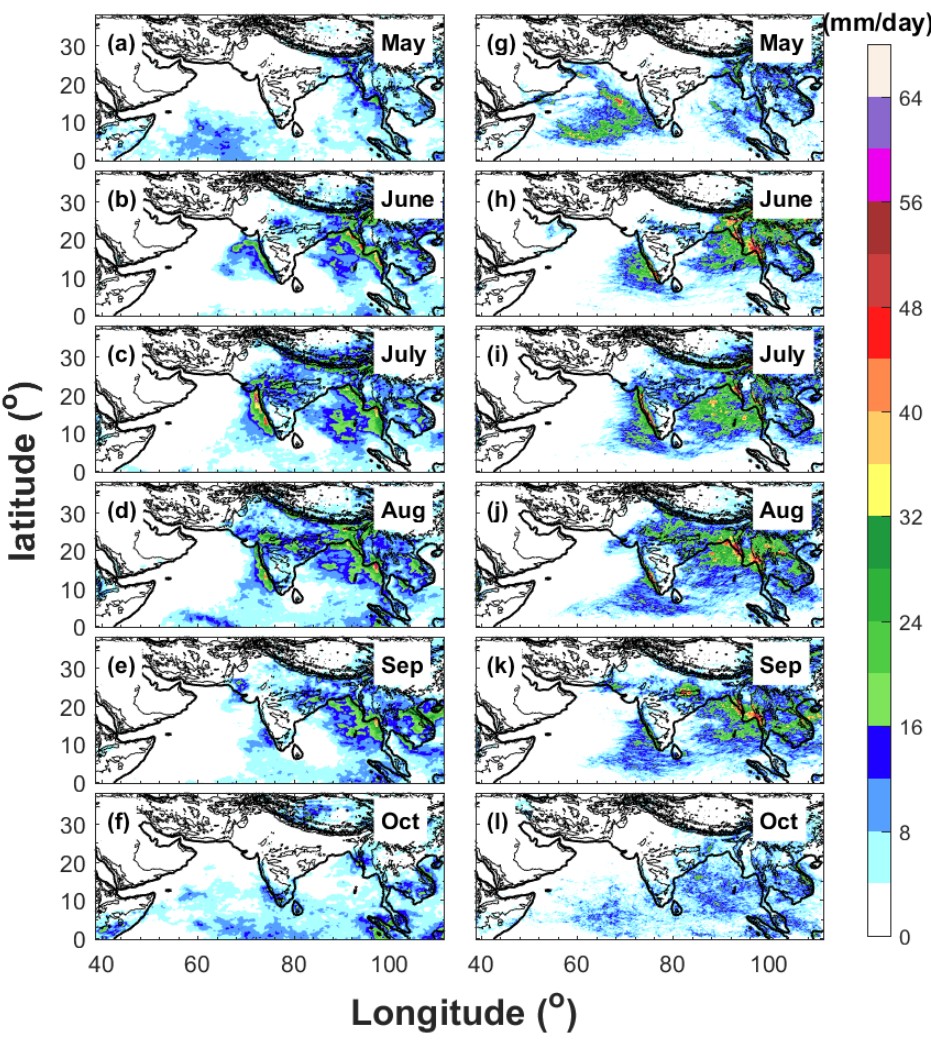

Figure 8. Spatial distributions of averaged daily surface precipitation from May to October in year 2011
derived from (a-f) TRMM and (g-l) WRF-gray.



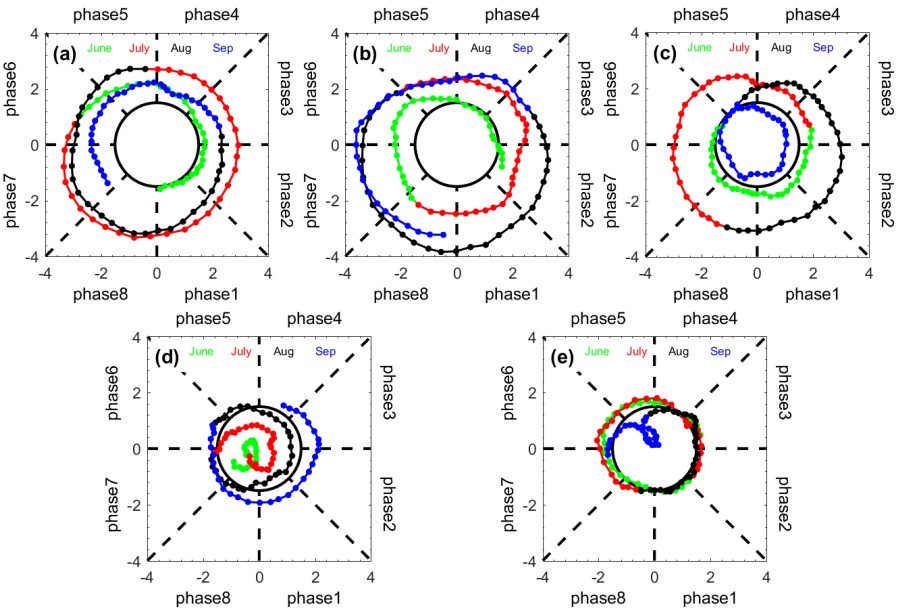

Figure 9. 2D phase space diagrams for the NLSA MISO indices. An anticlockwise propagation from the phase 1
represents MISO's northward propagation. The circle centered at the origin has radius equal to 1.5, which is
the threshold for identification of significant MISO events.





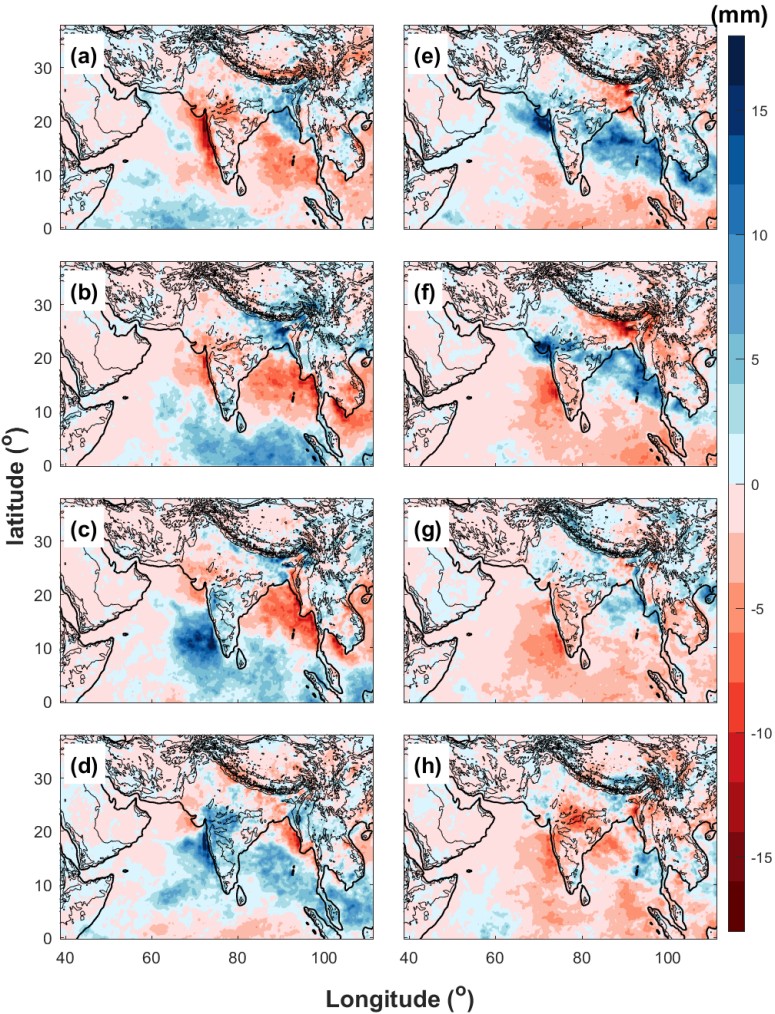

Figure 10. Phase composites of daily surface rainfall anomalies obtained from TRMM (Figure a-h: phase 1 to
775    8).



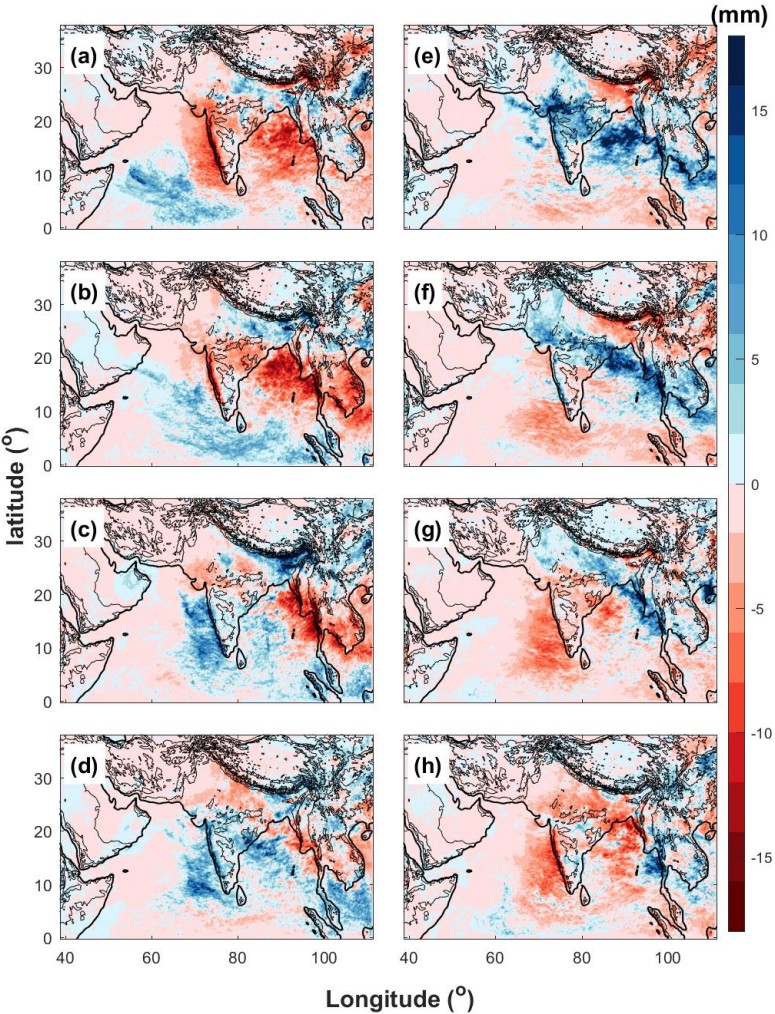

Figure 11. Phase composites of daily surface rainfall anomalies obtained from WRF-gray (Figure a-h: phase 1
to 8).





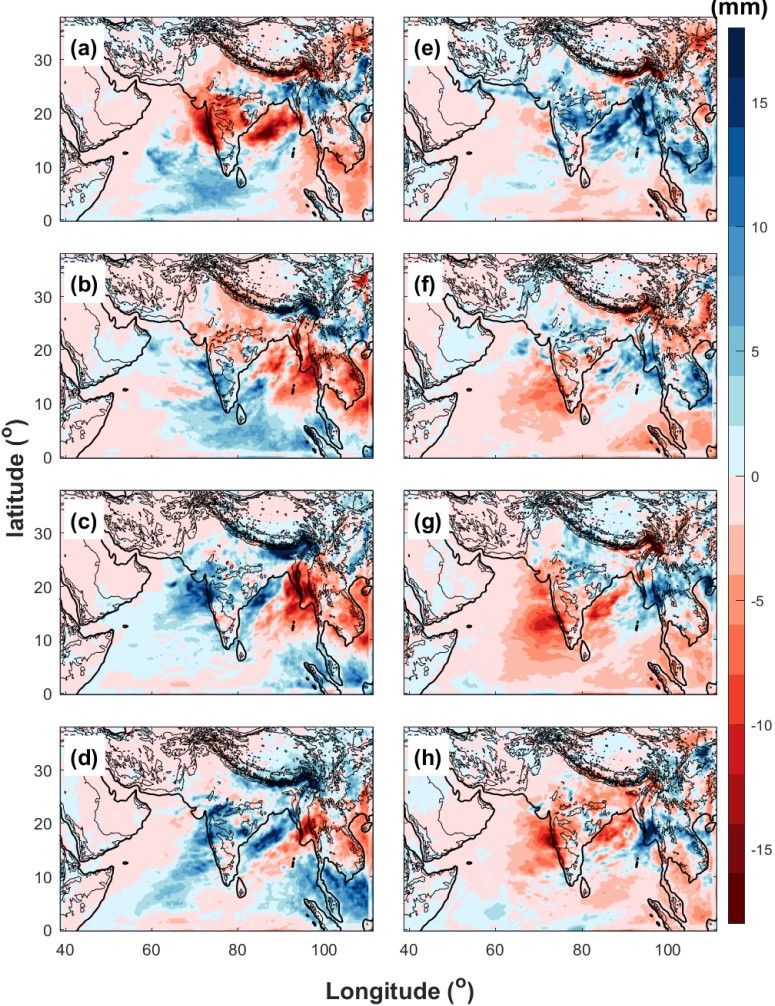

Figure 12. Phase composites of daily surface rainfall anomalies obtained from WRF-27km (Figure a-h: phase 1
to 8).





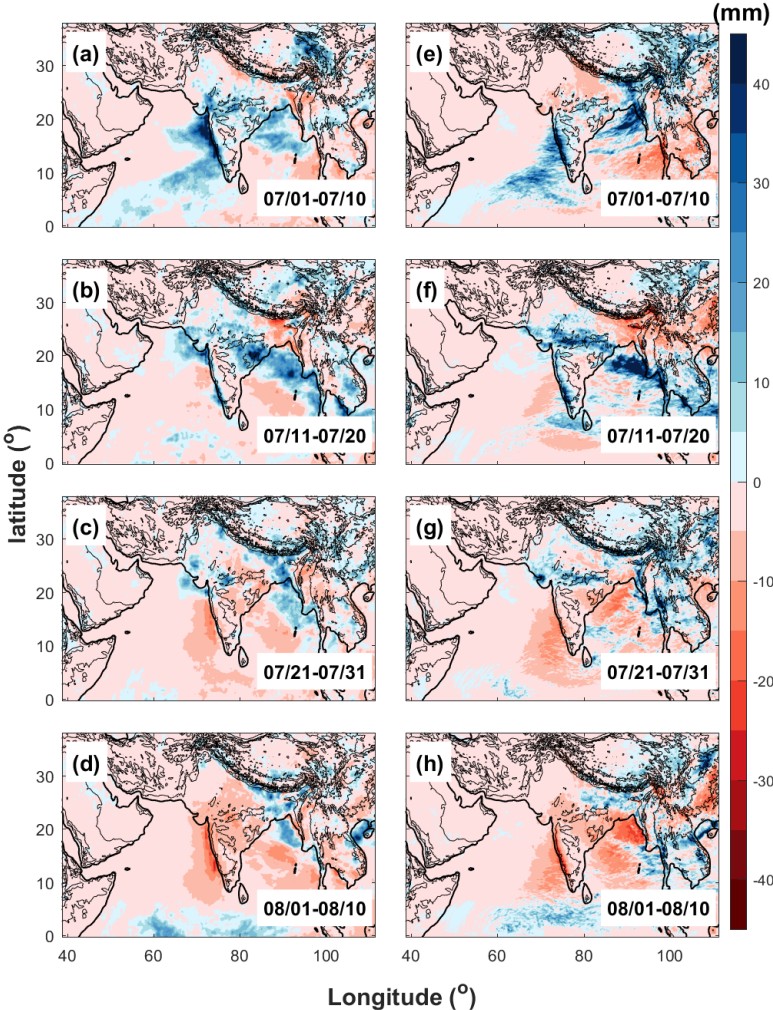

Figure 13. Spatial distributions of 10-day averaged daily surface rainfall anomalies in (a, e) 1-10 July, (d, f) 11-20 July, (c, g) 21-31 July and (d, h) 01-10 August, 2009 derived from TRMM (left panels) and WRF-gray (right panels).




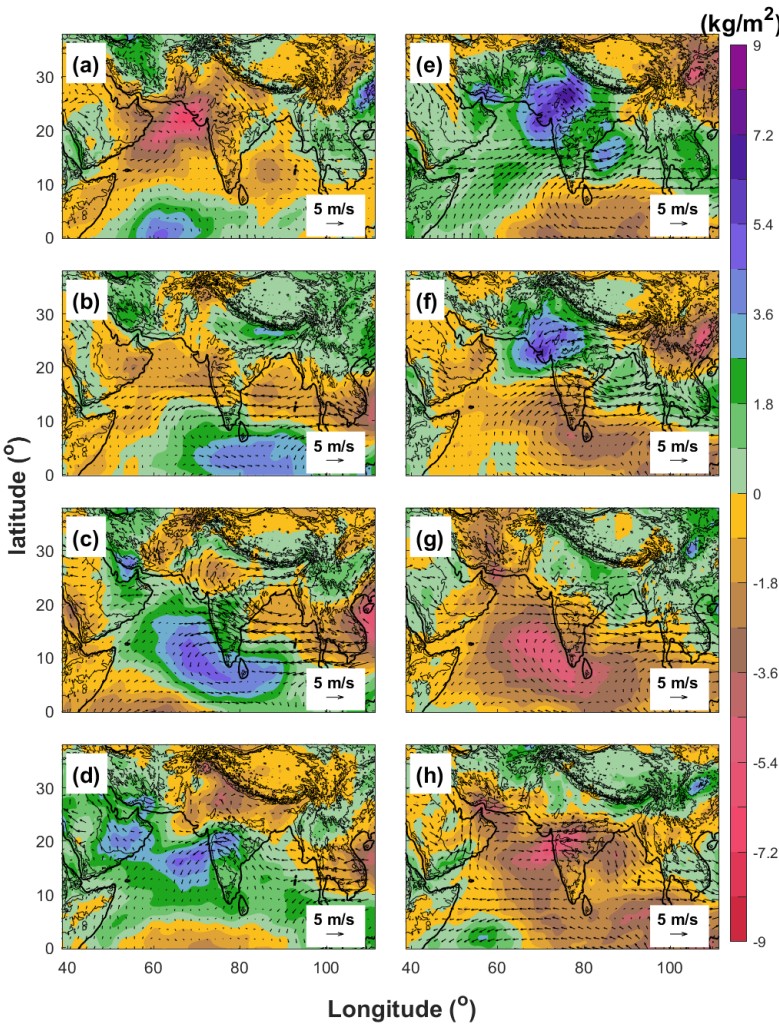

Figure 14. Phase composites of 850-hPa wind and precipitable water anomalies obtained from ERA-Interim
(Figure a-h: phase 1 to 8).





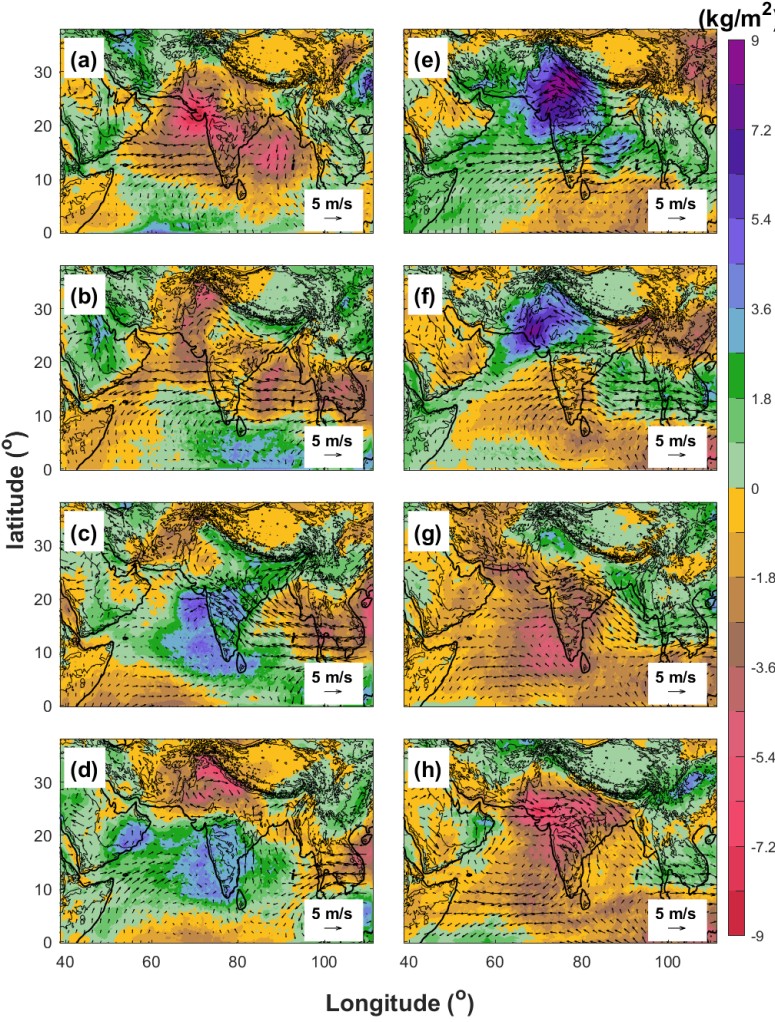

Figure 15. Phase composites of 850-hPa wind and precipitable water anomalies obtained from WRF-gray
(Figure a-h: phase 1 to 8).



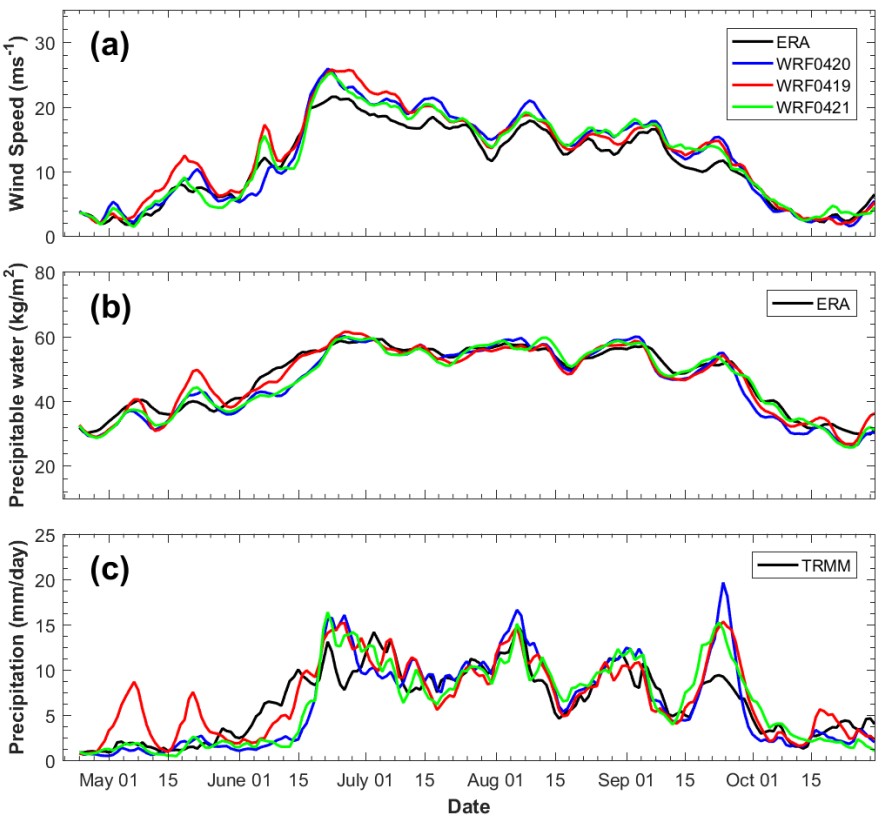

Figure 16. Temporal evolutions of (a) KELLF indices, (b) precipitable water averaged over the Indian
subcontinent and (c) daily surface precipitation averaged over the Indian subcontinent in year 2007 from
ERA-Interim/TRMM (black lines), WRF-gray simulation starts from April 20 (blue lines, control run), WRF-gray
simulation starts from April 19 (red lines) and WRF-gray simulation starts from April 21 (green lines).