# Peer review of "Regional Simulation of Indian summer Monsoon Intraseasonal Oscillations at Gray Zone Resolution"

_Atmospheric Chemistry and Physics, 2017_

## Referee Comment (RC1) · D. Straus (Referee) · 20 Aug 2017

This paper reports on simulation of the Indian Monsoon for five seasons with a regional climate model that has fine enough spatial resolution (9 km) to capture convection explicitly. The presentation of results in terms of monsoon break and active periods, and their relationship to the Monsoon Intraseasonal Oscillation, is quite thorough. Other than issues of grammar, the paper is quite readable and informative. There are a few concerns that should be addressed, however, before the paper is accepted in final form.

(1) More discussion should be given to the role of the boundary conditions (here from

[Figure]

ERA-Interim). These boundary conditions constrain the planetary scale flow to be close to analyses. Can the authors speculate how well a global model of this resolution would in simulating the Indian Monsoon, even in the mean?

(2) Related to (1), on lines 80-81 the authors state that: "Results show that GCMs are able to capture the fundamental features of the monsoon circulation reasonably well and also show some skills in reproducing the seasonal-averaged distributions of the monsoonal rainfall..." This may not be a valid generalization – this reviewer has the impression that some very high resolution global GCMs still have significant errors in the simulation of the mean Indian monsoon rainfall.

(3) Greater physical explanation of the MISO indices, based on "nonlinear Laplacian spectral analysis technique" should be given. One way in which this appears to differ from the multichannel singular spectral analysis (applied by, for example, Krishnamurthy and Shukla, 2007) is that the latter gives two distinct periods for MISO oscillations (45 and 20 days), while the method applied by the authors gives a single oscillation. Why the difference.

(4) What is the effect of the model top at 20 hPa and suppression of vertically propagating gravity waves?

(5) Figure 5 should show the seasonal mean Indian monsoon rainfall for each year, a single number for each year. This is referred to often, but not shown.

(6) The grammar of the article needs to be thoroughly checked and improved.

---

## Referee Comment (RC2) · Anonymous Referee #2 · 21 Aug 2017

The study investigates Indian summer monsoon and its intraseasonal variability from 2007 to 2011 using the WRF regional modeling system. Model evaluation against observational dataset shows that the model is able to simulate precipitation and large-scale circulation associated with the seasonal and intraseasonal variations of the monsoon system. In contrast to 27 km simulations with convective parameterization, the 9-km gray resolution simulations show significantly less bias in mean precipitation. The manuscript is presented with clarity. Figures are of high quality. I have a number of comments and suggestions.

General comments:

[Figure]

1. The main message of this study is that the gray resolution simulations have some advantages over coarse resolution simulations with cumulus parameterization. The manuscript attributes most of the difference to the use of the cumulus parameterization in the 27 km simulations. However, another potentially important factor is that topography is much better resolved in the 9 km simulations, which is potentially crucial for producing rain in the Himalaya foothills and Indian subcontinent (discussed several times in the manuscript). From reading the current manuscript it is unclear whether or not the 9-km simulations use the same topography as the 27 km simulations, and to what extent fine topography could have contributed to improvement. The revision may include some discussion on this issue.

2. The manuscript uses the MISO index constructed from the nonlinear Laplacian spectral analysis technique to evaluate model simulated MISOs. I suggest compute the MISO index from the model simulations (both the 27 km and 9 km simulations). This seems to be straightforward using projection of model data to the MISO mode. The benefit is that this would significantly simplify model evaluation and comparison between model simulations.

Specific comments:

Lines 74-77: This discussion is out of context. Why it is more complex than the MJO? Is there a reference for this?

Lines 84-85: a few GCMs can go down to 25 kilometers for TC (e.g., GFDL HiRAM)

Figures 4,5: The same quantities from the WRF-27 may also be included to show that the WRF-gray also improve temporal variations of circulation and rain in addition to the mean. Correlation coefficient may be computed to quantify the performance of simulations.

Figure 9: Year for each panel is missing in either title or caption.

Section 4.2 & Figure 10: The MISO index is computed with the GPCP precipitation, but

the phase composite is constructed with TRMM. There seems to be an inconsistency. Shouldn't the same rainfall observation data be used consistently?

Lines 120: 12 km is not considered as CRM resolution. Suggest change "CRM" to "cloud-permitting modeling"

Line 330: here GPCP is used, but the rest paper use TRMM. There seems to be some discrepancy.

Other technical comments:

Line 67: foothills

Line 72: in another world?

Line 337: delete "it"

Line 351: Phase -> phase

Line 371: Phase -> phase

Line 375: delete "should"

Line 377: what is "real one"? Need to be reworded.

Line 397: access -> assess

Line 443-447: this sentence is repetitive and unclear. Suggest rewrite this.

---

## Author Comment (AC1) · 30 Oct 2017

**Response to Reviewer #1**

**Comments:**

1. More discussion should be given to the role of the boundary conditions (here from ERA-Interim). These boundary conditions constrain the planetary scale flow to be close to analyses. Can the authors speculate how well a global model of this resolution would in simulating the Indian Monsoon, even in the mean?

**Thank you for your suggestion. The ERA-Interim reanalysis used as the boundary conditions in this manuscript is one of the latest global atmospheric reanalysis produced by the European Center for Medium-Range Weather Forecast (ECMWF). The reanalysis is produced with a sequential data assimilation scheme, advancing forward in time using 12-hourly analysis cycles. The zonal and meridional winds in the reanalysis are directly assimilated from observational data and extrapolate from locally observed parameters to unobserved parameters through 4D-Var scheme. Though the planetary scale flow in the reanalysis is not exactly the observation, it is verified well with the radiosondes and aircraft observations (Dee et al. 2011). So, the reanalysis dataset is the best estimate of spatially complete monsoon large scale flow we have and also be regarded as equivalent to observations by many users (Lin et al. 2014). Kishore et al. (2016) compared the Indian summer monsoon precipitation simulation in four different reanalysis datasets with the observations. The results show that the ERA-Interim has the highest skill to reproduce the Indian summer monsoon rainfall though obvious biases can still be found. Similar conclusions can also be found in Lin et al. (2014).**

**We have added the related discussion on Lines 174-180: The ERA-Interim reanalysis is produced with a sequential data assimilation scheme, advancing forward in time using 12-hourly analysis cycles. The zonal and meridional winds in the reanalysis are directly assimilated from observational data, thus the large scale monsoon circulation is well captured in the reanalysis. The Indian summer monsoon precipitation climatology in ERA-Interim has also been compared with that in other reanalysis datasets, and the results show that ERA-Interim has the highest skill to reproduce the Indian summer monsoon rainfall though obvious biases can still be found (Kishore et al., 2016; Lin et al., 2014).**

*Dee, D. P., and Coauthors, 2011: The Era-Interim Reanalysis: Configuration and Performance of the Data Assimilation System. Q. J. R. Meteorol. Soc., 137, 553-597.*

*Kishore, P., S. Jyothi, G. Basha, S. V. B. Rao, M. Rajeevan, I. Velicogna, and T. C. Sutterley, 2016: Precipitation Climatology over India: Validation with Observations and Reanalysis Datasets and Spatial Trends. Climate Dynam., 46, 541-556.*

*Lin, R., T. Zhou, and Y. Qian, 2014: Evaluation of Global Monsoon Precipitation Changes Based on Five Reanalysis Datasets. J. Climate, 27, 1271-1289.*

2. Related to (1), on lines 80-81 the authors state that: "Results show that GCMs are able to capture the fundamental features of the monsoon circulation reasonably well and also show

some skills in reproducing the seasonal-averaged distributions of the monsoonal rainfall…" This may not be a valid generalization – this reviewer has the impression that some very high resolutions global GCMs still have significant errors in the simulation of the mean Indian monsoon rainfall.

**We agree with the reviewer that the monsoon precipitation simulation is still a rigorous test for most GSMs, even for the seasonal rainfall climatology. While, some studies show that the spatial distribution of monsoon can be captured reasonably by some GCMs. We have improved the related statement on lines 78-80 to "Results show that GCMs are able to capture the fundamental features of the monsoon circulation reasonably well. However, the monsoon precipitation is still a rigorous test for most GCMs (e.g., Bhaskaran et al., 1995; Lau and Ploshay, 2009; Chen et al., 2011)."**

3. Greater physical explanation of the MISO indices, based on "nonlinear Laplacian spectral analysis technique" should be given. One way in which this appears to differ from the multichannel singular spectral analysis (applied by, for example, Krishnamurthy and Shukla, 2007) is that the latter gives two distinct periods for MISO oscillations (45 and 20 days), while the method applied by the authors gives a single oscillation. Why the difference.

**Thank you for your suggestion. As mentioned by the reviewer, there are two dominant intraseasonal oscillations in Krishnamurthy and Shukla (2007). The 45-day intraseasonal oscillation reflects the intraseasonal variation of large-scale monsoonal flows, like the movement of the monsoon trough. It is closely related to the Boreal Summer Intraseasonal Oscillation and generally known as the Monsoon Intraseasonal Oscillation (MISO, Suhas et al., 2013; Ajayamohan and Goswami, 2003). MISO is a northeastward propagating mode. It is still a big challenge for most current global model to well simulate this northeastward propagating mode (Ajayamohan et al., 2014). However, the 20-day intraseasonal oscillation is a westward propagating mode, which represents the propagation and life cycle of monsoon lows and depressions embedded in the monsoon trough. The MISO indices we used in the manuscript based on the developed nonlinear Laplacian spectral analysis (Sabeerali et al. 2017) representing the full life cycle of the northeastward propagating mode with 30-60 days periodicity (MISO), which is similar with the 45-day oscillation in Krishnamurthy and Shukla (2007).**

**Per the reviewer's suggestion, we have improved the statement related to the MISO indices and added more discussions on the differences between NLSA and the classical covariance-based approaches (EEOF or MSSA) on line 348-362: "NLSA is a nonlinear data analysis technique that combines ideas from kernel methods for harmonic analysis, delay embedding of dynamical systems and machine learning (Belkin and Niyogi, 2003; Packard et al., 1980; Sauer et al., 1991; Coifman and Lafon, 2006). Compared to the classical covariance-based approaches (for example Suhas et al., 2013; Krishnamurthy and Shukla, 2007), a key advantage of NLSA is that it is able to extract the spatiotemporal modes of variability spanning multiple timescales without requiring bandpass filtering or seasonal partitioning of the input data. Compared to the MISO indices based on the extend EOF (EEOF) and multichannel singular spectral analysis (MSSA), the NLSA-based MISO indices have improved timescale separation, higher memory and higher predictability. The MISO indices constructed by NLSA can better resolve the temporal and spatial characteristics of the MISO. For example, the NLSA-based**

**MISO indices have better temporal phase coherence while maintaining the isolating ability of MISO from broad band dataset. It can better resolve the tilted structure of MISO convection and the associated atmospheric circulation pattern through phase composites and also explain more fractional variance over the ocean regions (Sabeerali et al., 2017)."**

4. What is the effect of the model top at 20 hPa and suppression of vertically propagating gravity waves?

**We have added the related statement on lines 150-153: "The high model top (~ 27 km) allows the development of deep convection, especially over the mountainous area. While the sponge layer above the tropopause can damp the gravity waves produced by the deep convective activity or steep terrain and prevent the upward propagating gravity-wave energy be bounced back to the troposphere."**

5. Figure 5 should show the seasonal mean Indian monsoon rainfall for each year, a single number for each year. This is referred to often, but not shown.

**Thank you for the suggestion. The seasonal mean Indian monsoon rainfall in TRMM observations, WRF-gray and WRF-27km have been shown in Figure 5 the related statement has also been added on lines 276-280.**

6. The grammar of the article needs to be thoroughly checked and improved.

**Thanks. We have thoroughly checked and corrected the grammar mistake in this manuscript.**

**Response to Reviewer #2**

**Comments:**

1. The main message of this study is that the gray resolution simulations have some advantages over coarse resolution simulations with cumulus parameterization. The manuscript attributes most of the difference to the use of the cumulus parameterization in the 27 km simulations. However, another potentially important factor is that topography is much better resolved in the 9 km simulations, which is potentially crucial for producing rain in the Himalaya foothills and Indian subcontinent (discussed several times in the manuscript). From reading the current manuscript it is unclear whether or not the 9-km simulations use the same topography as the 27 km simulations, and to what extent fine topography could have contributed to improvement. The revision may include some discussion on this issue.

**Thank you for your suggestion. The original input topography data of the WRF simulations is at 30s (~0.9 km) resolution. After interpolated to the model grid, the topography is better resolved in the 9 km simulations than in the 27 km simulations (9km versus 27km). We have added the related discussions on lines 206-216: "An apparent moist bias of surface precipitation can be found for all 5 years (2007 to 2011) in WRF-27km, while this systematic bias is reduced considerably in WRF-gray. One reason of the moist bias reduction is the topography is better resolved in WRF-gray than that in WRF-27km (the spatial resolution of topography is 3-times higher in WRF-gray). Thus, the local convective activity is better simulated and the total surface precipitation is less over the Himalaya foothills and West Ghats in WRF-gray (not shown here). Besides the mountainous area, surface rainfall simulation over the plain and oceanic regions also shows strong moist bias in WRF-27km and it is improved dramatically in WRF-gray. It shows that the higher model resolution and better simulation of large-scale atmospheric circulation also have considerable contributes to the improvement of monsoon rainfall simulation in WRF-gray, which will be discussed in details in the following sections."**

2. The manuscript uses the MISO index constructed from the nonlinear Laplacian spectral analysis technique to evaluate model simulated MISOs. I suggest compute the MISO index from the model simulations (both the 27 km and 9 km simulations). This seems to be straightforward using projection of model data to the MISO mode. The benefit is that this would significantly simplify model evaluation and comparison between model simulations.

**Thank you. Per the MISO indices definition in Sabeerali et al. (2017), in order to identify the modes northward propagating boreal summer MISO, the nonlinear Laplacian spectral analysis technique need to be applied to the long-term continuous rainfall data (19980101-20141231), yields a hierarchy of Laplace-Beltrami Eigen functions capturing coherent patterns of rainfall variability, then extract the MISO purely from annual and Semiannual cycles and higher frequency oscillations. While, for our current study, only 5 boreal summers are simulated from**

2007 to 2011, which cannot be used as a continuous dataset to identify the MISO indices same as that in Sabeerali et al. (2017). To keep the consistency with the MISO indices definition in Sabeerali et al. (2017), we decide still use the long-term rainfall observational dataset to define the MISO indices in the current study.

Per your suggestion, we used the TRMM observations instead of GPCP data in the revision to calculate the NLSA-based MISO indices to keep the consistency of the discussion. All related figures are changed (Figs 09, 10, 11, 12, 14 and 15) and the related statements are also improved. Though the values of the MISO indices have little changes because of the rainfall intensities in GPCP and TEMM data are different, the phases and interannual variations of the MISO indices derived from two datasets are almost same. The phase composites of daily surface rainfall anomalies are also similar, which shows the results are robust to the choice of rainfall data to produce the MISO indices.

**Specific comments:**

1. Lines 74-77: This discussion is out of context. Why it is more complex than the MJO? Is there a reference for this?

We have reworded the statement to: "The MISO is influenced by a number of physical processes (Goswami, 1994). Its interactions with the mean monsoon circulation and other tropical oscillations make its propagating characteristics very complex (Krishnamurthy and Shukla, 2007)."

2. Lines 84-85: a few GCMs can go down to 25 kilometers for TC (e.g., GFDL HiRAM)

Thank you for your comment. We have added the related statement on lines 84-85: "though a few GCMs can go down to 25 km for tropical cyclones forecast, like GFDL HiRAM".

3. Figures 4,5: The same quantities from the WRF-27 may also be included to show that the WRF-gray also improve temporal variations of circulation and rain in addition to the mean. Correlation coefficient may be computed to quantify the performance of simulations.

Thank you for your suggestion. We have included WRF-27km in Figures 4 and 5. The related statements are also added on lines 269-273: "However, the strength of Somali Jet in WRF-27km is much stronger when compared to WRF-gray and ERA-Interim. It is one of the reasons why there is a high moist bias of monsoon rainfall in WRF-27km. In general, the simulation at gray zone resolution can better capture the large scale atmospheric circulation of Indian summer monsoon than the coarser resolution simulation with cumulus scheme." and on lines 295-304: "Compared to WRF-gray, the surface rainfall in WRF-27km shows a high moist bias over the Indian subcontinent, which is consistent with the overprediction of the Somali Jet strength shown in Fig. 4 and the analysis of seasonal mean rainfall in Fig. 2. Also, the intraseasonal variation of monsoon rainfall is poorer simulated in WRF-27km when compared with that in WRF-gray. For example, the "active" and "break" spells from August to September in 2007 are well captured by WRF-gray while not simulated in WRF-27km. The 5-years averaged correlation coefficient of precipitation between WRF-gray (WRF-27km) and TRMM observations is 0.786 (0.714). These results show that the simulation at gray zone resolution can better capture both the mean intensity and the intraseasonal variation of Indian summer monsoon than the

**coarser resolution simulation using cumulus scheme.”**

4. Figure 9: Year for each panel is missing in either title or caption.
   **Thank you. The related statement has been added in the caption of Figure 9.**

5. Section 4.2 & Figure 10: The MISO index is computed with the GPCP precipitation, but the phase composite is constructed with TRMM. There seems to be an inconsistency. Shouldn't the same rainfall observation data be used consistently?
   **We have calculated the new MISO indices using TRMM observations. Figures 09, 10, 11, 12, 14 and 15 are changed based on the new MISO indices and related statements in sections 4.1 and 4.2 are also improved.**

6. Lines 120: 12 km is not considered as CRM resolution. Suggest change "CRM" to "cloud-permitting modeling"
   **Changed to "cloud-permitting model"**

7. Line 330: here GPCP is used, but the rest paper use TRMM. There seems to be some discrepancy.
   **We have calculated the new MISO indices using TRMM observations. Figures 09, 10, 11, 12, 14 and 15 are changed based on the new MISO indices and related statements in sections 4.1 and 4.2 are also improved.**

**Other technical comments:**
1. Line 67: foothills
   **Done**

2. Line 72: in another world?
   **Changed to "in another word"**

3. Line 337: delete "it"
   **Done**

4. Line 351: Phase -> phase
   **Done**

5. Line 371: Phase -> phase
   **Done**

6. Line 375: delete "should"
   **Done**

7. Line 377: what is "real one"? Need to be reworded.
   **Reworded to "which reflects that the model simulated MISO is stronger than that in the satellite observations."**

8. Line 397: access -> assess

**Done**

9. Line 443-447: this sentence is repetitive and unclear. Suggest rewrite this.

**Reworded to: "These biases may induced by various reasons such as the choices of surface scheme, the model domain size and the initial conditions which the dynamical systems are highly sensitive to."**

---

## Referee Report (RR1)

Minor Concerns:

(1) Lines 355-356: "NLSA-based MISO indices have...higher memory and higher predictability." This should be briefly explained.

(2) Lines 521-522. Having the model simulate a single tropical cyclone over the Arabian Sea which did not occur in nature is not a model bias. A model bias would refer to the tendency of the model to produce significantly more or fewer cyclones than in nature, but that apparently is not the case. That the model can actually generate its own tropical cyclone is probably a strength of the model.

(3) Lines 538-541. The role of initial conditions is not well expressed. The fact that perturbations in the initial conditions lead to changes in the characteristics of the ISO and MISO may say something real about their predictability. The model bias is by definition an expression of the model error averaged over many initial conditions.

Grammar:
Line 59: "...to the agricultural and industrial productions..."
should be "on agricultural and industrial production..."

Line 66: "...and characterized..." should be "...and is characterized ...."

Lines71-72: " ... also has considerable influences on the onset and withdrawal of the ISM, which, in another word, determining the length of the rainy season" should be "...also have a considerable influence on the onset and withdrawal of the ISM. In other words, these MISO phases determine the length of the rainy season"

Line 84 "lager" should be "larger"

Line 85 "...for tropical cyclones forecast" should be "for tropical cyclone forecasts"

Line 87: "which also leads"  should be "leading to "

Line 89: "the way" should be "one way"

Line 114: "and affects" should be "which affect"

Line 134: "avoid using" should be "avoiding the use of"

Line 151: "While the sponge layer" should be "The sponge layer"
Line 152  "and prevent the upward propagating gravity-wave energy be bounced back to the troposphere" should be: "and prevents upward propagating gravity-wave energy from being reflectected back to the troposphere"

Line 166: "than YSU scheme" should be "than the YSU scheme"
Line 167: " is still deserve" should be "still deserves"
Line 168: "quantifiactions" should be "quantification"

Line 193 "similar with WRF-gray except" should be "to that of the WRF-gray configuration, except that a"

Line 213: "It shows that" should be "These results show that"
Line 214: "better simulation of" should be "a better simulation of the"
Lines 214-215: "also have have considerable" should be cut.
Lines 215-216: "which will be discussed in details" should be "a finding that will be discussed in detail"

Line 251: "that driving" should be "that drive"
Line 270: "It is" should be "This is"
Line 280: "most drought" should be "driest"
Line 283: "which are associated" should be "changes which are associated"
Line 293: "5-yr ISMs" should be cut.
Line 296: "which is" should be cut.
Line 298: "is poorer simulated" should be "is more poorly simulated"
Line 306: "Similar with" should be "Similar to"
Line 310: "Bay of Bangle" should be "Bay of Bengal"
Line 329: "with that in" should be "to that in"
Line 330: "reaches" should be "reaching"
Line 346: "developed" should be cut.
Line 374: "which is" should be cut.
Line 377: "amplitude" should be "amplitudes"
Lines 382-384: "(subtracted the mean daily rainfall of 5-yr monsoon seasons) obtained from TRMM observation based on the NLSA MISO indices. The phase composites are computed by averaging the significant MISO activities in each phase space occurred in the 5-yr monsoon seasons" should be:
"obtained from TRMM observations based on the NLSA MISO indices. (The daily climatology used to define the anomalies is based on the 5-years.) The phase composites are computed by averaging significant MISO anomalies."
Line 390: "tropic ocean" should be "tropical ocean"
Line 391: "West Ghats" should be "the western Ghats"
Line 394: "form" should be "forms"
Line 396: "northeast" should be "the northeast"
Line 398: "west coast" should be "the west coast"
Line 404: "which shows" should be ", showing "
"statistic reflects" should be "statistics reflect"
Line 432: "Besides the phase composite, the evolutions" should be "The evolution"
Line 434-435: "10-day evolutions of rainfall" should be: "The evolution of rainfall"
Line 442: "most area" should be "most areas"
Line 444: "negative with rainfall over" should be "negative while rainfall over"
Line 462: "which transports more" should be "transporting more"
Line 468: "lead to negative precipitation water" should be "leading to negative precipitable water"

Line 501: "Model domain" should be "the model domain"
Line 511: "is also stronger" should be "are also stronger:

---

## Author Response (AR2)

**Minor Concerns:**

(1) Lines 355-356: "NLSA-based MISO indices have...higher memory and higher predictability." This should be briefly explained.

**Thank you for your suggestion. More explanation has been added: "The hindcasts of these indices using operational extended-range output show that NLSA-based MISO indices remain predictable out to ~ 3 weeks while other indices only have ~2 weeks predictability."**

(2) Lines 521-522. Having the model simulate a single tropical cyclone over the Arabian Sea which did not occur in nature is not a model bias. A model bias would refer to the tendency of the model to produce significantly more or fewer cyclones than in nature, but that apparently is not the case. That the model can actually generate its own tropical cyclone is probably strength of the model.

**Thank you for your advice. We agree with you and have improved the statement as: "Results show that WRF-gray could reproduce the spatial patterns of the monthly rainfall in each year and well capture the monsoon rainfall centers over West Ghats, central India, Himalaya foothills and the west coast of Myanmar. However, biases of rainfall intensity and position can still be found in WRF-gray."**

(3) Lines 538-541. The role of initial conditions is not well expressed. The fact that perturbations in the initial conditions lead to changes in the characteristics of the ISO and MISO may say something real about their predictability. The model bias is by definition an expression of the model error averaged over many initial conditions.

**Thank you for your suggestion. The related statement has been improved as: "The ISM simulation at gray zone resolution is also sensitive to its initial conditions. More comprehensive investigation on the predictability of the ISO and MISO in RCM at gray zone resolution is deserved future studies."**

**Grammar:**

(1) Line 59: "...to the agricultural and industrial productions..." should be "on agricultural and industrial production..."
**Done**

(2) Line 66: "...and characterized..." should be "...and is characterized ...."
**Done**

(3) Lines71-72: " ... also has considerable influences on the onset and withdrawal of the ISM, which, in another word, determining the length of the rainy season" should be "...also have a considerable influence on the onset and withdrawal of the ISM. In other words, these MISO phases determine the length of the rainy season"
**Done**

(4) Line 84 "lager" should be "larger"
**Done**

(5) Line 85 "...for tropical cyclones forecast" should be "for tropical cyclone forecasts"
**Done**

(6) Line 87: "which also leads" should be "leading to "
**Done**

(7) Line 89: "the way" should be "one way"
**Done**

(8) Line 114: "and affects" should be "which affect"
**Done**

(9) Line 134: "avoid using" should be "avoiding the use of"
**Done**

(10) Line 151: "While the sponge layer" should be "The sponge layer"
**Done**

(11) Line 152 "and prevent the upward propagating gravity-wave energy be bounced back to the troposphere" should be: "and prevents upward propagating gravity-wave energy from being reflected back to the troposphere"
**Done**

(12) Line 166: "than YSU scheme" should be "than the YSU scheme"
**Done**

(13) Line 167: " is still deserve" should be "still deserves"
**Done**

(14) Line 168: "quantifiactions" should be "quantification"
**Done**

(15) Line 193 "similar with WRF-gray except" should be "to that of the WRF-gray configuration, except that a"
**Done**

(16) Line 213: "It shows that" should be "These results show that"
**Done**

(17) Line 214: "better simulation of" should be "a better simulation of the"
**Done**

(18) Lines 214-215: "also have considerable" should be cut.
**Done**

(19) Lines 215-216: "which will be discussed in details" should be "a finding that will be discussed in detail"
**Done**

(20) Line 251: "that driving" should be "that drive"
**Done**

(21) Line 270: "It is" should be "This is"
**Done**

(22) Line 280: "most drought" should be "driest"
**Done**

(23) Line 283: "which are associated" should be "changes which are associated"
**Done**

(24) Line 293: "5-yr ISMs" should be cut.
**Done**

(25) Line 296: "which is" should be cut.
**Done**

(26) Line 298: "is poorer simulated" should be "is more poorly simulated"
**Done**

(27) Line 306: "Similar with" should be "Similar to"
**Done**

(28) Line 310: "Bay of Bangle" should be "Bay of Bengal"
**Done**

(29) Line 329: "with that in" should be "to that in"
**Done**

(30) Line 330: "reaches" should be "reaching"
**Done**

(31) Line 346: "developed" should be cut.
**Done**

(32) Line 374: "which is" should be cut.
**Done**

(33) Line 377: "amplitude" should be "amplitudes"
**Done**

(34) Lines 382-384: "(subtracted the mean daily rainfall of 5-yr monsoon seasons) obtained from TRMM observation based on the NLSA MISO indices. The phase composites are computed by averaging the significant MISO activities in each phase space occurred in the 5-yr monsoon seasons" should be: "obtained from TRMM observations based on the NLSA MISO indices. (The daily climatology used to define the anomalies is based on the 5-years.) The phase composites are computed by averaging significant MISO anomalies."
**Done**

(35) Line 390: "tropic ocean" should be "tropical ocean"
**Done**

(36) Line 391: "West Ghats" should be "the western Ghats"
**Done**

(37) Line 394: "form" should be "forms"
**Done**

(38) Line 396: "northeast" should be "the northeast"
**Done**

(39) Line 398: "west coast" should be "the west coast"
**Done**

(40) Line 404: "which shows" should be ", showing ". "statistic reflects" should be "statistics reflect"
**Done**

(41) Line 432: "Besides the phase composite, the evolutions" should be "The evolution"
**Done**

(42) Line 434-435: "10-day evolutions of rainfall" should be: "The evolution of rainfall"
**Done**

(43) Line 442: "most area" should be "most areas"
**Done**

(44) Line 444: "negative with rainfall over" should be "negative while rainfall over"
**Done**

(45) Line 462: "which transports more" should be "transporting more"
**Done**

(46) Line 468: "lead to negative precipitation water" should be "leading to negative precipitable water"

**Done**

(47) Line 501: "Model domain" should be "the model domain"

**Done**

(48) Line 511: "is also stronger" should be "are also stronger"

**Done**